# GATED DOMAIN UNITS FOR MULTI-SOURCE DOMAIN GENERALIZATION

## ABSTRACT

Distribution shift (DS) is a common problem that deteriorates the performance of learning machines. To tackle this problem, we postulate that real-world distributions are composed of elementary distributions that remain invariant across different environments. We call this an *invariant elementary distribution* (I.E.D.) assumption. The I.E.D. assumption implies an invariant structure in the solution space that enables knowledge transfer to unseen domains. To exploit this property in domain generalization (DG), we developed a modular neural network layer that consists of Gated Domain Units (GDUs). Each GDU learns an embedding of an individual elementary distribution that allows us to encode the domain similarities during the training. During inference, the GDUs compute similarities between an observation and each of the corresponding elementary distributions which are then used to form a weighted ensemble of learning machines. Because our layer is trained with backpropagation, it can naturally be integrated into existing deep learning frameworks. Our evaluation on image, text, graph, and time-series data shows a significant improvement in the performance on out-of-training target domains without domain information and any access to data from the target domains. This finding supports the practicality of the I.E.D. assumption and demonstrates that our GDUs can learn to represent these elementary distributions.

## 1 INTRODUCTION

A fundamental assumption in machine learning is that training and test data are independently and identically distributed (I.I.D.). This assumption ensures consistency-results from statistical learning theory, meaning that the learning machine obtained from an empirical risk minimization (ERM) attains the lowest achievable risk as sample size grows (Vapnik, 1998; Schölkopf, 2019). Unfortunately, a considerable amount of research and real-world applications in the past decades has provided a staggering evidence against this assumption (Zhao et al., 2018; 2020; Ren et al., 2019; Taori et al., 2020) (see D'Amour et al. (2020) for case studies). The violation of the I.I.D. assumption is usually caused by a distribution shift (DS) and can result in inconsistent learning machines (Sugiyama & Kawanabe, 2012), implying the loss of performance guarantee of machine learning models in the real world. Therefore, to tackle DS, recent work advocates for domain generalization (DG) (Blanchard et al., 2011; Muandet et al., 2013; Li et al., 2017; 2018b; Zhou et al., 2021a). This generalization to utterly unseen domains is crucial for robust deployment of the models in practice, especially when new, unforeseeable domains emerge after model deployment. However, the most important question that DG seeks to answer is how to identify the right *invariance* that allows for generalization.

The contribution of this work is twofold. First, we advocate that real-world distributions are composed of smaller "units" called *invariant elementary distributions* that remain invariant across different domains; see Section 2.1. Second, we propose to implement this hypothesis through so-called gated domain units (GDUs). Specifically, we developed a modular neural network layer that consists of GDUs. Each GDU learns an embedding of an individual elementary domain that allows us to express the domain similarities during training. For this purpose, we adopt the theoretical framework of reproducing kernel Hilbert space (RKHS) to retrieve a geometrical representation of each distribution in the form of a kernel mean embedding (KME) without information loss (Berlinet & Thomas-Agnan, 2004; Smola et al., 2007; Sriperumbudur et al., 2010; Muandet et al., 2017). This representation accommodates methods based on analytical geometry to measure similarities between distributions.

We show that these similarity measures can be learned and utilized to improve the generalization capability of deep learning models to previously unseen domains.

The remainder of this paper is organized as follows: Our theoretical framework is laid out in Section 2 with our modular DG layer implementation shown in Section 3. In Section 4, we outline related work. Our experimental evaluations are presented in Section 5. Finally, we discuss potential limitations of our approach and future work in Section 6.

## 2 DOMAIN GENERALIZATION WITH INVARIANT ELEMENTARY DISTRIBUTIONS

We assume a mixture component shift for the multi-source DG setting. This shift refers to the most common DS stating that the data is made up of different sources, each with its own characteristics, and their proportions vary between the training and test scenario (Quinonero-Candela et al., 2022). Our work thus differs in the assumption from related work in DG, in which the central assumption is the covariate shift (i.e., the conditional distribution of the source and test data stays the same) (David et al., 2010). In the following, let $\mathcal{X}$ and $\mathcal{Y}$ be the input and output space, with a joint distribution $\mathbb{P}$. We are given a set of $D$ labeled source datasets $\{\mathcal{D}_i^s\}_{i=1}^D$ with $\mathcal{D}_i^s \subseteq \mathcal{X} \times \mathcal{Y}$. Each of the source datasets is assumed to be I.I.D. generated by a joint distribution $\mathbb{P}_i^s$ with support on $\mathcal{X} \times \mathcal{Y}$, henceforth denoted *domain*. The set of probability measures with support on $\mathcal{X} \times \mathcal{Y}$ is denoted by $\mathcal{P}$. The multi-source dataset $\mathcal{D}^s$ comprises the merged individual source datasets $\{\mathcal{D}_j^s\}_{j=1}^D$. We aim to minimize the empirical risk, see Section 3.3 for details. Important notation is summarized in Table 1.

### 2.1 INVARIANT ELEMENTARY DISTRIBUTIONS

Similar to Mansour et al. (2009; 2012); Hoffman et al. (2018a), we assume that the distribution of the source dataset can be described as a convex combination $\mathbb{P}^s = \sum_{j=1}^D \alpha_j^s \mathbb{P}_j^s$ where $\alpha^s = (\alpha_1^s, \ldots, \alpha_D^s)$ is an element of the probability simplex, i.e., $\alpha^s \in \Delta^D := \{\alpha \in \mathbb{R}^D \mid \alpha_j \geq 0 \wedge \sum_{j=1}^D \alpha_j = 1\}$. In other words, $\alpha_j$ quantifies the contribution of each individual source domain to the combined source domain.

In contrast, we generalize their problem descriptions: We express the distribution of each domain as a convex combination of $K$ elementary distributions $\{\mathbb{P}_j\}_{j=1}^K \subset \mathcal{P}$, meaning that $\mathbb{P}^s = \sum_{j=1}^K \alpha_j \mathbb{P}_j$ where $\alpha \in \Delta^K$. Our main assumption is that *these elementary distributions remain invariant across the domains*. The advantage is that we can find an invariant subspace at a more elementary level, as opposed to when we consider the source domains as some sort of basis for all unseen domain. Figure 1 illustrates this idea.

Table 1: Important notation

| | |
|---|---|
| K | number of elementary distributions |
| M | number of elementary domain bases |
| N | number of basis vectors |
| $\mathbb{P}^s$ | combined multi-source distribution |
| $\mathbb{P}_j^s$ | $j$-th single-source distribution |
| $\mathbb{P}_j$ | $j$-th elementary distribution |
| $V_j$ | $j$-th domain basis |
| $v_k^j$ | $k$-th vector in $V_j$ |
| $\alpha_j^s$ | coefficient for $\mathbb{P}_j^s$ |
| $\alpha_j$ | coefficient for $\mathbb{P}_j$ |
| $\beta_{ij}$ | coefficient for sample $x_i$ and $\mu_{V_j}$ |

Theoretically speaking, the I.E.D assumption is appealing because it implies the invariant structure in the solution space, as shown in the following lemma. The proof is given in Appendix A.1.

**Lemma 1.** *Let $\mathcal{L} : \mathcal{Y} \times \mathcal{Y} \to \mathbb{R}_+$ be a non-negative loss function, $\mathcal{F}$ a hypothesis space of functions $f : \mathcal{X} \to \mathcal{Y}$, and $\mathbb{P}^s(X, Y)$ a data distribution. Suppose that the I.E.D assumption holds, i.e., there exist $K$ elementary distributions $\mathbb{P}_1, \ldots, \mathbb{P}_K$ such that any data distribution can be expressed as $\mathbb{P}^s(X, Y) = \sum_{j=1}^K \alpha_j \mathbb{P}_j(X, Y)$ for some $\alpha \in \Delta^K$. Then, the corresponding Bayes predictor $f^* \in \arg\min_{f \in \mathcal{F}} \mathbb{E}_{(X,Y) \sim \mathbb{P}}[\mathcal{L}(Y, f(X))]$ is Pareto-optimal with respect to a vector of elementary risk functionals $(R_1, \ldots, R_K) : \mathcal{F} \to \mathbb{R}_+^K$ where $R_j(f) := \mathbb{E}_{(X,Y) \sim \mathbb{P}_j}[\mathcal{L}(Y, f(X))]$.*

Lemma 1 implies that, under the I.E.D assumption, Bayes predictors must belong to a subspace of $\mathcal{F}$ called the Pareto set $\mathcal{F}_{\text{Pareto}} \subset \mathcal{F}$ which consists of Pareto-optimal models. The model $f$ is said to be Pareto-optimal if there exists no $g \in \mathcal{F}$ such that $R_j(g) \geq R_j(f)$ for all $j \in \{1, \ldots, K\}$ with $R_j(g) > R_j(f)$ for some $j$; see, e.g., Sener & Koltun (2018, Definition 1). In other words, the I.E.D assumption allows us to translate the invariance property of data distributions to the solution space. Since Bayes predictors of *all* future test domains must lie within the Pareto set, which is a

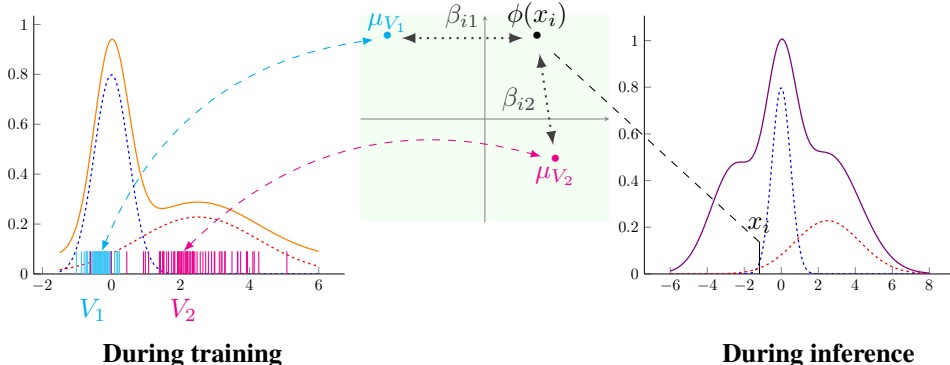

Figure 1: A visualization of an "invariant elementary distribution (I.E.D.)" assumption for domain generalization (DG): the observed data distributions (orange and violet) are composed of the same set of *unobserved* elementary distributions (blue and red) that remain invariant across different domains. Hence, the first challenge during the training phase (left panel) is to extract these elementary distributions from the observed data (orange). The unobserved elementary distributions are represented by the elementary bases $V_1$ and $V_2$ (cyan and pink). The second challenge during the inference phase (right panel) is to create a weighted ensemble of learning machines that utilize the similarities between the embedding of the unseen observation $\phi(x_i)$ and the embeddings of these distributions $\mu_{V_1}$ and $\mu_{V_2}$ in the RKHS $\mathcal{H}$ (green rectangle) as weights $\beta_{i1}$ and $\beta_{i2}$.

strict subset of the original hypothesis space, it is still possible to identify the optimal predictors of future test domains, even without additional data from the test domains, except the I.E.D. assumption itself. Hence, given data from the training domains, it is sufficient for the purpose of generalization to maintain only solutions within this Pareto set during the training time.

Unfortunately, neither the elementary distributions nor the weights $\boldsymbol{\alpha}$ are known in practice. Motivated by this theoretical insight, our DG layer presented in Section 3 is designed to uncover them from a multi-source training dataset $\mathcal{D}^s$. While Lemma 1 shows the theoretical appeal of the I.E.D. assumption, we discuss below a situation in which it might hold in practice. The limitations will be discussed later in Section 6.

**Real-world example.** In this work, we postulate that the elementary domain bases are the invariant subspaces that allow us to generalize to unseen domains. In practice, the question arises if and when elementary domains evolve. Consider that we aim to learn to predict the risk of developing Diabetes from laboratory data from Europe and then infer the risk from data from the United States of America. Naturally, factors influencing the data-generating process may change, such as the level of physical activity and nutritional habits. While, to a certain degree, these common factors remain invariant across continents, each of these factors' contributions may differ. In terms of our assumptions, we model each of these factors with a corresponding elementary distribution $\mathbb{P}_j$. For a previously unseen individual, we can then determine the coefficients $\alpha_j^s$ and quantify each factor's contribution without any information about the individual's origin.

## 2.2 KERNEL MEAN EMBEDDING OF DISTRIBUTIONS

We leverage the KME of distributions (Berlinet & Thomas-Agnan, 2004; Smola et al., 2007; Muandet et al., 2017) to discover the elementary distributions and evaluate similarities between them. Let $\mathcal{H}$ be a reproducing kernel Hilbert space (RKHS) of real-valued functions on $\mathcal{X}$ with a reproducing kernel $k : \mathcal{X} \times \mathcal{X} \to \mathbb{R}$ (Schölkopf et al., 2001). The KME of a probability measure $\mathbb{P} \in \mathcal{P}$ in the RKHS $\mathcal{H}$ is defined by a mapping $\phi(\mathbb{P}) = \mu_{\mathbb{P}} := \int_{\mathcal{X}} k(\mathbf{x}, \cdot) \, d\mathbb{P}(\mathbf{x})$. We assume that the kernel $k$ is characteristic, i.e., the mapping $\mu_{\mathbb{P}}$ is injective (Fukumizu et al., 2004; Sriperumbudur et al., 2008). Theoretically, this essential assumption ensures that there is no information loss when mapping the distribution into $\mathcal{H}$. Given the samples $\{x_1, \ldots, x_n\}$ generated I.I.D. from $\mathbb{P}$, $\mu_{\mathbb{P}}$ can be approximated by the empirical KME $\hat{\mu}_{\mathbb{P}} = (1/n) \sum_{i=1}^{n} k(x_i, \cdot) = (1/n) \sum_{i=1}^{n} \phi(x_i)$. We refer non-expert readers to Muandet et al. (2017) for a thorough review on this topic.

**Challenges.** Figure 1 depicts two challenges that come with our assumption of elementary distributions. First, since we do not have access to the samples from the hidden elementary distributions, the elementary KME cannot be estimated directly from the samples at hand. To overcome this challenge, we instead seek a proxy KME $\mu_{V_j} := (1/N) \sum_{k=1}^{N} \phi(v_k^j) = (1/N) \sum_{k=1}^{N} k(v_k^j, \cdot)$ for each elementary KME $\mu_{\mathbb{P}_j}$ from a domain basis $V_j$, where $V_j = \{v_1^j, \ldots, v_N^j\} \subseteq \mathcal{X}$ for all $j \in \{1, \ldots, M\}$. Hence, the KME $\mu_{V_j}$ can be interpreted as the KME of the empirical probability measure $\hat{\mathbb{P}}_{V_j} = (1/N) \sum_{k=1}^{N} \delta_{v_k^j}$. Here, we assume that $M = K$. The sets $V_j$ are referred to as *elementary domain basis*. Intuitively, the elementary domain basis $V_1, \ldots, V_M$ represents each elementary distribution by a set of vectors that mimic samples generated from the corresponding distribution. In Figure 1, $V_1$ and $V_2$ as well as their mapping in $\mathcal{H}$ visualize this first challenge.

The second challenge is the objective of learning the unknown similarity between a single sample $x_i$ and an elementary domain $V_j$, which we denote by $\beta_{ij}$. Considering the advantage of KMEs, that is to tackle this challenge from a geometrical viewpoint, we quantify similarities between KMEs. For example, in Figure 1, the similarity between $\phi(x_i)$ and $\mu_{V_1}$ ($\beta_{i1}$) and $\mu_{V_2}$ ($\beta_{i2}$) could be quantified as their distance or angle. These similarity coefficients enable our *Domain Generalization Layer* to represent a convex combination of elementary domain-specific learning machines, commonly known as ensembles. We introduce our layer in the following Section 3.

## 3 DOMAIN GENERALIZATION LAYER

This section aims to transfer the theoretical ideas presented in Section 2 into a deep learning framework. For the purpose of implementation, let $x \in \mathbb{R}^{h \times w}$ denote the input data point and $h_\xi : \mathbb{R}^{h \times w} \to \mathbb{R}^e$ the feature extractor (FE) that maps the input into a low-dimensional representation $\tilde{x} \in \mathbb{R}^e$. Then the prediction layer $g_\theta : \mathbb{R}^e \to \mathcal{Y}$ infers the label $y$. To tackle the DG problem, we introduce a layer module called the gated domain unit (GDU). A GDU consists of three main components: (1) a similarity function $\gamma : \mathcal{H} \times \mathcal{H} \to \mathbb{R}$ that is the same for all elementary domains, (2) an elementary basis $V_j$ and (3) a learning machine $f(\tilde{x}, \theta_j)$ for each elementary domain $j \in \{1, \ldots, M\}$. The architecture of the layer proposed herein is depicted in Figure 2.

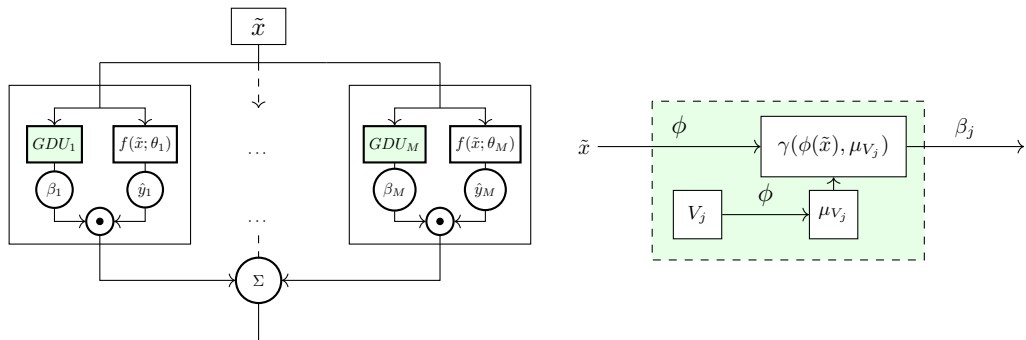

Figure 2: Visualization of the DG layer (left panel) and its main component, the GDU (right panel). The DG layer consists of several GDUs that represent the elementary distributions. During training, these GDUs learn the elementary domain bases $V_1, \ldots, V_M$ that approximate these distributions.

Essentially, the process is as follows: First, the $j$-th GDU takes $\tilde{x}_i$ as an input and yields $\beta_{ij}$ as an output. The KME of each domain basis $V_j$ is required in order to apply $\gamma$ to compute similarity between $\tilde{x}_i$ and $V_j$. These KMEs are obtained by $\phi(V_j) := \mu_{V_j} = (1/N) \sum_{k=1}^{N} \phi(v_k^j) = (1/N) \sum_{k=1}^{N} k(v_k^j, \cdot)$. The GDU, therefore, has the task to allocate coefficients $\beta_{ij}$ for each elementary domain based on a similarity function $\gamma$. The function $\gamma$ outputs the $\beta_{ij} = \gamma(\phi(\tilde{x}_i), \mu_{V_j})$ coefficients that in turn represent similarities between the KME of both, the corresponding domain basis $V_j$ and the input $\tilde{x}_i$. Theoretically speaking, $\mu_{V_j}$ and the feature mapping $\phi(\tilde{x}_i)$ are elements of the associated RKHS $\mathcal{H}$, which allow us to evaluate similarities of non-linear features in a higher dimensional feature space. Each GDU is then connected to a learning machine $f(\tilde{x}_i, \theta_j)$ that yields an elementary domain-specific inference. The final prediction of the layer is then an ensemble of

these learning machines $g_\theta(\tilde{x}_i) = \sum_{j=1}^M \beta_{ij} f(\tilde{x}_i, \theta_j)$ where $\theta = (\theta_1, \ldots, \theta_M)$. In Figure 2, we give an overview of how data is processed and information is stored in the GDU.

In summary, GDUs leverage the invariant elementary distribution (I.E.D.) assumption and represent our algorithmic contribution: The elementary domain bases are stored as weights in the layer. Storing information as a weight matrix (i.e., domain memory) allows to learn the elementary domain bases efficiently using backpropagation. Hence, we avoid the dependency on problem-adaptive methods (e.g., domain-adversarial training) and domain information (e.g., domain labels).

### 3.1 DOMAIN SIMILARITY MEASURES

For the similarity function $\gamma$, we consider two similarity measures $H(\phi(\tilde{x}), \mu_{V_j})$, namely the cosine similarity (CS) (Kim et al., 2019) and maximum mean discrepancy (MMD) (Borgwardt et al., 2006; Gretton et al., 2012). To ensure that the resulting coefficients $\beta_i$ lie on the probability simplex, we apply the kernel softmax function (Gao et al., 2019) and interpret its output as the similarity between an observation $\tilde{x}$ and an elementary domain basis $V_i$. We get

$$\beta_{ij} = \gamma(\phi(\tilde{x}_i), \mu_{V_j}) = \frac{\exp\big(\kappa H(\phi(\tilde{x}_i), \mu_{V_j})\big)}{\sum_{k=1}^M \exp\big(\kappa H(\phi(\tilde{x}_i), \mu_{V_k})\big)}, \tag{1}$$

where $\kappa > 0$ is a positive softness parameter for the kernel softmax. Geometrically speaking, these similarities correspond to the angle and distance of two KMEs in the RKHS $\mathcal{H}$. The function $\phi$ maps the observation $\tilde{x}$ and domain basis $V_j$ into $\mathcal{H}$ meaning that $\phi(\tilde{x}) = \mu_{\delta_{\tilde{x}}} = k(\tilde{x}, \cdot)$ is the KME of a Dirac measure $\delta_{\tilde{x}}$ and $\phi(V_j) = \mu_{V_j} = (1/N) \sum_{k=1}^N k(v_k^j, \cdot)$.

**CS.** The CS function $H(\phi(\tilde{x}_i), \mu_{V_j}) = \frac{\langle \phi(\tilde{x}_i), \mu_{V_j} \rangle_\mathcal{H}}{\|\phi(\tilde{x}_i)\|_\mathcal{H} \|\mu_{V_j}\|_\mathcal{H}}$ is used as an angle-based similarity.

**MMD.** We consider the MMD for calculating a distance-based similarity measure. The distance is then given as $\|\phi(\tilde{x}_i) - \mu_{V_j}\|_\mathcal{H}$. Subsequently, the similarity function $H$ is the negative MMD: $H(\phi(\tilde{x}_i), \mu_{V_j}) = -\|\phi(\tilde{x}_i) - \mu_{V_j}\|_\mathcal{H}$. The intuition behind the negative MMD is to put higher weights on samples that are closer to the KME of an elementary domain basis.

### 3.2 PROJECTION-BASED GENERALIZATION

For classification tasks, we introduce an alternative approach to infer the $\beta_i$ coefficients that is based on the idea of kernel sparse coding (Gao et al., 2010; 2013). Herein the goal is to find an approximated representation of each feature mapping $\phi(\tilde{x}_i)$ using the elements of a dictionary $\{\mu_{V_j}\}_{j=1}^M$. This approach allows us to approximate the feature mapping with these elements by $\phi(\tilde{x}_i) \approx \sum_{j=1}^M \beta_{ij} \mu_{V_j}$. In contrast to the aforementioned approaches, an elementary domain KME $\mu_{V_j}$ does not necessarily represent the KME of an elementary distribution $\mu_{\mathbb{P}_j}$. Therefore, we present another approach that aims to find a set $\{\mu_{V_j}\}_{j=1}^M$ that permits $\mu_{\mathbb{P}^s}$ to be represented as a linear combination.

Since $\mathbb{P}$ is assumed to be a convex combination of elementary distributions, we can find a linear combination to represent $\mu_{\mathbb{P}^s}$ by the domain KMEs $\mu_{V_j}$, as long as $\mu_{\mathbb{P}^s} \in \mathcal{H}_M := \text{span}\{\mu_{V_j} \mid j = 1, \ldots, M\}$. The RKHS $\mathcal{H}_M$ is a subspace of the actual RKHS $\mathcal{H}$, which allows us to represent elements of $\mathcal{H}$ at least approximately in the subspace $\mathcal{H}_M$. By keeping the $\mathcal{H}_M$ large, we gain more representative power. To make $\mathcal{H}_M$ as large as possible, we have to ensure its spanning elements are linearly independent or, even better, orthogonal. Orthogonal KMEs ensure two desirable properties. First, pairwise orthogonal elements in $\mathcal{H}_M$ guarantee no redundancy. Second, having orthogonal elements allows us to make use of the orthogonal projection. This projection geometrically yields the best approximation of $\phi(\tilde{x})$ in $\mathcal{H}_M$. In other words, we can achieve the best possible approximation of the feature mapping by using its orthogonal components (see Proposition 3.1). The orthogonal projection is given by

$$\Pi_{\mathcal{H}_M} : \mathcal{H} \to \mathcal{H}_M, \quad \phi(\tilde{x}) \mapsto \sum_{i=1}^M \frac{\langle \phi(\tilde{x}), \mu_{V_j} \rangle_\mathcal{H}}{\|\mu_{V_j}\|_\mathcal{H}^2} \mu_{V_j}. \tag{2}$$

**Proposition 3.1.** *For a KME $\mu_\mathbb{P}$ of a given mixture distribution $\mathbb{P}$ the following holds $\mu_\mathbb{P} \in span\{\mu_{V_j} \mid V_j, \forall j = 1, \ldots, M\}$, where $\langle \mu_{V_i}, \mu_{V_j} \rangle_\mathcal{H} = 0, \forall i \neq j$ (i.e., the KME of the elementary domains basis are pairwise orthogonal). The value of the function $\sum_{j=1}^{M} \|\mu_\mathbb{P} - \beta_j \mu_{V_j}\|^2_{\mathcal{H}_k}$ is minimal if the coefficients are set as $\beta_j^* = \langle \mu_\mathbb{P}, \mu_{V_j} \rangle_\mathcal{H} / \|\mu_{V_j}\|^2_\mathcal{H}$.*

The Proposition 3.1 can be used to give an approximation of $\mu_\mathbb{P}$ by projecting it into $\mathcal{H}_M$, i.e., $\mu_\mathbb{P} \approx \sum_{j=1}^{M} \beta_j \mu_{V_j}$, where $\beta_j = \langle \mu_\mathbb{P}, \mu_{V_j} \rangle_\mathcal{H} / \|\mu_{V_j}\|^2_\mathcal{H}$. This best approximation property is the main advantage of our assumption in Proposition 3.1 (i.e., having orthogonal KME) and thus a potential advantage of projection-based DG. Appendix A.2 provides the proof of Proposition 3.1.

## 3.3 Model Training

For model training, we adapt the domain adaptation (DA) framework from Zhuang et al. (2021). Thus, our learning objective function is formalized as $\mathcal{L}(g) + \lambda_D \Omega_D(\|g\|_\mathcal{H})$. The goal of the training can be described in terms of the two components of this function. Consider a batch of training data $\{x_1, \ldots, x_b\}$, where $b$ is the batch size. During training, we minimize the loss function $\mathcal{L}(g) = \frac{1}{b} \sum_{i=1}^{b} \mathcal{L}(\hat{y}_i, y_i) = \frac{1}{b} \sum_{i=1}^{b} \mathcal{L}(\sum_{j=1}^{M} \gamma(\phi(\tilde{x}_i), \mu_{V_j}) f_j(\tilde{x}_i), y_i)$ for an underlying task and the respective batch size. In addition, our objective is that the model learns to distinguish between different domains. Thus, the regularization $\Omega_D$ is introduced to control the domain basis. In our case, we require the regularization $\Omega_D$ to ensure that the KMEs of the elementary domain basis are able to represent the KMEs of the elementary domains. Therefore, we minimize the MMD between the feature mappings $\phi(\tilde{x}_i)$ and the associated representation $\sum_{j=1}^{M} \beta_{ij} \mu_{V_j}$. Note that $\beta_{ij} = \gamma(\phi(\tilde{x}_i), \mu_{V_j})$. Hence, the regularization $\Omega_D = \Omega_D^{OLS}$ is defined as $\Omega_D^{OLS}(\|g\|_\mathcal{H}) = \frac{1}{b} \sum_{i=1}^{b} \|\phi(\tilde{x}_i) - \sum_{j=1}^{M} \beta_{ij} \mu_{V_j}\|^2_\mathcal{H}$ (see Appendix B.2 for details). The intuition is the objective to represent each feature mapping $\phi(\tilde{x}_i)$ by the domain KMEs $\mu_{V_j}$. Thus, we try to minimize the MMD between the feature map and a combination of $\mu_{V_j}$. The minimum of the stated regularization can be interpreted as the ordinary least square-solution of a regression-problem of $\phi(\tilde{x}_i)$ by the components of $\mathcal{H}_M$. In other words, we want to ensure that the basis $V_j$ is contained in feature mappings $\phi(\tilde{x}_i)$.

In the particular case of projection, we want the KME of the elementary domain to be orthogonal to ensure high expressive power. For this purpose, the additional term $\Omega_D^\perp$ will be introduced to ensure the desired orthogonality. Considering a kernel function with $k(x, x) = 1$, orthogonality would require the Gram matrix $K_{ij} = \langle \mu_{V_i}, \mu_{V_j} \rangle_\mathcal{H}$ to be close to the identity matrix $I$. There are a variety of methods for regularizing matrices available (Xie et al., 2017; Bansal et al., 2018). A well-known method to ensure orthogonality is the soft orthogonality (SO) regularization $\Omega_D^\perp = \lambda \|K - I\|^2_F$ (Bansal et al., 2018). As pointed out by Bansal et al. (2018), the spectral restricted isometry property (SRIP) and mutual coherence (MC) regularization can be a promising alternative for SO and thus are additionally implemented in the DG layer. Hence, in the case of projection, the regularization is given by $\Omega_D(\|g\|_\mathcal{H}) = \lambda_{OLS} \Omega_D^{OLS}(\|g\|_\mathcal{H}) + \lambda_{ORTH} \Omega_D^\perp(\|g\|_\mathcal{H})$, $\lambda_{OLS}, \lambda_{ORTH} \geq 0$.

Lastly, sparse coding is an efficient technique to find the least possible basis to recover the data subject to a reconstruction error (Olshausen & Field, 1997). Several such applications yield strong performances, for example in the field of computer vision (Lee et al., 2007; Yang et al., 2009). Kernel sparse coding transfers the reconstruction problem of sparse coding into $\mathcal{H}$ by using the mapping $\phi$, and, by applying a kernel function, the reconstruction error is quantified as the inner product (Gao et al., 2010; 2013). To ensure sparsity, we apply the $L_1$-norm on the coefficients $\beta$ and add $\Omega_D^{L_1}(\|\gamma\|) := \|\gamma(\phi(\tilde{x}_i), \mu_{V_j})\|_1$ to the regularization term $\Omega_D$ with the corresponding coefficient $\lambda_{L_1}$. Appendix B.3 gives a visual overview of the model training.

## 4 Related Work

DG, also known as out-of-distribution (OOD) generalization, is among the hardest problems in machine learning (Blanchard et al., 2011; Muandet et al., 2013; Arjovsky et al., 2019). In contrast, DA, which predates DG and OOD problems, deals with a slightly simpler scenario in which some data from the test distribution are available (Ganin et al., 2015). Hence, based on the available data, the task is to develop learning machines that transfer knowledge learned in a source domain specifically to the target domain. Approaches pursued in DA can be grouped primarily into (1) discrepancy-based

DA (Sun et al., 2016; Peng & Saenko, 2018; Ben-David et al., 2010; Fang et al., 2020; Tzeng et al., 2014; Long et al., 2015; Baktashmotlagh et al., 2016) (2) adversary-based DA (Tzeng et al., 2017; Liu & Tuzel, 2016; Ganin et al., 2015; Long et al., 2018), and (3) reconstruction-based DA (Bousmalis et al., 2016; Hoffman et al., 2018b; Kim et al., 2017; Yi et al., 2017; Zhu et al., 2017; Ghifary et al., 2014). In DA, learning the domain-invariant components requires access to unlabeled data from the target domain. Unlike problems in DA, where the observed data from the test domains can be used to find the most appropriate invariant structures (Ben-David et al., 2010), the lack thereof in DG calls for a postulation of invariant structure that will enable the OOD generalization.

To enable generalization to unseen domains without any access to data from them, researchers have made significant progress in the past decade and developed a broad spectrum of methodologies (Zhou et al., 2021a;c; Li et al., 2019; Blanchard et al., 2011). For thorough review see, e.g., Zhou et al. (2021a); Wang et al. (2021). Existing works can be categorized into methods based on domain-invariant representation learning (Muandet et al., 2013; Li et al., 2018b;d), meta-learning (Li et al., 2018a; Balaji et al., 2018), data augmentation (Zhou et al., 2020), to name a few. Another recent stream of research from a causal perspective includes invariant risk minimization (Arjovsky et al., 2019), invariant causal prediction (Peters et al., 2016), and causal representation learning (Schölkopf et al., 2021). The overall motivation here is to learn the representation that is robust to domain-specific spurious correlations. In other words, it is postulated that "causal" features are the right kind of invariance that will enable OOD generalization. Despite the successful applications, DG remains a challenging research gap.

We differentiate our work from existing ones as follow. First, we postulate the existence of domain-invariant structure at the distributional level rather than at the data representation, which is a common assumption in DG. This is motivated by theoretical results (Mansour et al., 2009; Hoffman et al., 2018a), stating that a distribution-weighted combination of source hypotheses represents the ideal hypothesis. Furthermore, our distributional assumption, as we argued in Section 2, generalizes previous work that proposes to use domain-specific knowledge to tackle the problem of DG from a more elementary setting. For example, approaches such as Piratla et al. (2020); Monteiro et al. (2021) can be compared to our GDUs as domain-specific predictors, in the special case, where each elementary domain represents a single source domain. However, GDUs do not assume the existence of a single common classifier for all the domains, providing a combination of multiple common classifiers shared between different source domains.

Second, we incorporate the I.E.D. assumption directly into our model's architecture, as shown in Figure 2. Designing effective architectures for DG has been largely neglected (Zhou et al., 2020, Sec. 4.1). Last, we do not assume access to domain information. Although obtaining such information can be difficult in practice, see our short discussion in Appendix C.4 (Niu et al., 2017), DG methods that can deal with their absence (e.g., Huang et al. (2020); Carlucci et al. (2019); Li et al. (2018c)) are yet scarce (Zhou et al., 2020, Sec. 4.2).

## 5 EXPERIMENTS

Since ERM is one of the strongest baselines in DG (Gulrajani & Lopez-Paz, 2020; Koh et al., 2021), we, first, compare our approach compared to ERM and ensemble learning (Table 2 and Appendix C.1). Second, we benchmark our approach to state-of-the-art DG (e.g., CORAL, LISA, IRM, FISH, Group DRO) methods focusing on image, graph, and text data (Table 3 and Appendix C.4). Third, we analyse the GDUs robustness gainst DS that occurs in daily clinical practice (Table 12 and Appendix C.3). Finally, in Appendix C.2, we conduct an ablation study focusing on the representation learned during training (Appendix C.2.2). In our experiments, we distinguish two modes of training the DG layer: fine tuning (FT), where we extract features using a pre-trained model, and end-to-end training (E2E), where the FE and the DG layer are jointly trained[1].

### 5.1 PROOF-OF-CONCEPT BASED ON DIGITS CLASSIFICATION

Following Feng et al. (2020) among others, we create a multi-source dataset by combining five publicly available digits image datasets, namely MNIST (Lecun et al., 1998), MNIST-M (Ganin & Lempitsky, 2015), SVHN (Netzer et al., 2011), USPS, and Synthetic Digits (SYN) (Ganin &

---

[1]All source code is made available on GitHub.

Lempitsky, 2015). The task is to classify digits between zero and nine. Each of these datasets is considered an out-of-training target domain which is inaccessible during training, and the remaining four are the source domains. Details are given in Appendix C.1. Table 2 summarizes the results for the most challenging out-of-training target domain, namely MNIST-M. In Appendix C.1, we provide the results on the remaining target domains in Table 7 and a discussion heuristics for choosing hyperparameters for our GDUs. Our method noticeably improves for all datasets mean accuracy and decreases the standard deviation in comparison to the ERM and ensemble baselines, making the results more stable across the ten iterations reported.

We also compare our methods with related work that uses domain information and data augmentation, based on the results of Li et al. (2021) (Table 9 in Appendix C.1). Although data augmentation in DG is a comparatively strong approach and, at the same time, we do not use domain information; we obtain comparable results to the baselines reported by Li et al. (2021).

**Ablation study**    We chose the digits dataset to analyze each component of our DG layer in 1st paragraph in Appendix C.1 and C.2. We (A) vary $M$, $N$ on Figure 9, and the strength of the regularization terms on Figure 6, Figure 7, and Figure 8 to assess the sensitivity of the DG layer to the choice of hyperparameters, (B) visualize the output of the FE (Figure 11). Our ablation study in (A) reveals stable results across different sets of hyper-parameters. While the layer is not sensitive to the choice of regularization strength, we recommend not to omit the regularization completely, although the computational expenses decrease without the orthogonal regularization. As an illustration in (B), we project the output of the FE trained with a dense layer (ERM) and with the DG layer by t-SNE (t-distributed stochastic neighbor embedding). The GDU-trained FE yields more concentrated and bounded clusters in comparison to the one trained by ERM. Hence, we observe a positive effect on the representation learned by the FE.

Table 2: Results Digits experiment. The mean (standard deviation) accuracy for ten runs is reported. Best results are **bold**.

|       |            | MNIST-M          |
|-------|------------|------------------|
| *ERM* | *Single*   | *63.00 (3.20)*   |
|       | *Ensemble* | *62.87 (1.50)*   |
| **FT**| CS         | 68.55 (0.80)     |
|       | MMD        | 68.62 (0.70)     |
|       | PROJECTION | 68.56 (0.91)     |
| **E2E**| CS        | **69.25 (0.61)** |
|       | MMD        | 69.04 (0.83)     |
|       | PROJECTION | 68.67 (0.98)     |

### 5.2  WILDS BENCHMARK

To challenge the I.E.D. assumption and the OOD generalization capabilities of the GDUs, we use WILDS, a curated set of real-world experiments for benchmarking DG methods (Koh et al., 2021). Further, WILDS is a semi-synthetic dataset set that operates under similiar assumptions as the source component shift (Koh et al., 2021). We consider the following eight datasets: *Camelyon17*, *FMoW*, *FMoW*, *Amazon*, *iWildCam*, and *RxRx1*, *OGB-MolPCBA*, *Civil-Comments*, and *PovertyMap*, which represent the task of real-world DG. We closely follow Koh et al. (2021) for the experiments. Details on datasets and benchmark methods are given in Appendix C.4. We present our benchmarking in Table 3. Our results are achieved *out-of-the-box* (i.e., default parameters) since hyperparameter optimization has a substantial impact on the generalization performance (Gulrajani & Lopez-Paz, 2020), and we aim to highlight the improvements solely attributable to our GDUs.

First, we observe the strengths and weaknesses of the benchmarks in the different data sets, all of which are lower than ERM at least once. In contrast, although GDUs show similar behavior across the datasets, performing very well for some datasets (e.g., FMoW, Poverty Map), they, however, do not fall below ERM across all GDU experiments conducted. In addition, the baselines require domain information. Our approach requires less information, yet, achieving comparable results to the benchmarks.

### 5.3  ECG EXPERIMENT

The PhysioNet/Computing in Cardiology Challenge 2020 (Perez Alday et al., 2021; Goldberger et al., 2000; Perez Alday et al., 2020) aims to identify clinical diagnoses from 12-lead ECG recordings from 6 different databases. This publicly available pooled dataset contains 43,101 recordings sampled with various sampling frequencies and lengths. Each recording is labeled as having one or more of 24 cardiac abnormalities; hence, the task is to perform a multi-label binary classification. For our experiment, we iterate over the databases, taking one at a time as the test domain while utilizing

Table 3: Results on WILDS benchmarking tasks. Our results are achieved *out-of-the-box* (i.e., default parameters) without hyperparameter optimization. We use a grey background to highlight methods using no domain information for DG. We compute the metrics following Koh et al. (2021) and report the mean (standard deviation). Best benchmark and GDU results are **bold**.

| | | CAMELYON17 | FMoW | AMAZON | iWILDCAM | RxRx1 | OGB-MolPCBA | CIVIL COMMENTS | POVERTY MAP |
|---|---|---|---|---|---|---|---|---|---|
| | | AVG ACC | WORST-REG ACC | 10% ACC | MACRO F1 | AVG ACC | AVG WORST-REGION ACC | WORST-GROUP ACC | WORST-U/R R |
| ERM | | 70.3 (6.4) | 31.3 (0.17) | 53.8 (0.8) | 31.0 (1.3) | 29.9 (0.4) | **27.2 (0.3)** | 56.0 (3.6) | 0.45 (0.06) |
| ERM ENSEMBLE | | 70.0 (9.4) | 34.3 (0.18) | 54.0 (0.5) | 29.5 (0.4) | 29.6 (0.3) | 26.9 (0.3) | 55.6 (0.8) | **0.52 (0.08)** |
| CORAL | | 59.3 (7.7) | 32.8 (0.66) | 52.9 (0.8) | **32.8 (0.1)** | 28.4 (0.3) | 17.9 (0.5) | 65.6 (1.3) | 0.44 (0.07) |
| FISH | | 74.7 (7.1) | **34.6 (0.18)** | 53.3 (0.0) | 22.0 (1.8) | - | - | **75.3 (0.6)** | - |
| IRM | | 64.2 (8.1) | 32.8 (2.09) | 52.4 (0.8) | 15.1 (4.9) | 8.2 (1.1) | 15.6 (0.3) | 64.2 (8.1) | 0.43 (0.07) |
| GROUP DRO | | 68.4 (7.3) | 31.1 (1.66) | 53.3 (0.0) | 23.9 (2.1) | 22.5 (0.3) | 22.4 (0.6) | 68.4 (7.3) | 0.39 (0.06) |
| LISA | | **77.1 (6.9)** | 35.5 (0.81) | **54.7 (0.0)** | - | **31.9 (1.0)** | - | 72.9 (1.0) | - |
| CGD | | 69.4 (7.9) | 32.0 (2.26) | - | - | - | - | 69.1 (1.9) | 0.43 (0.04) |
| ARM-BN | | - | 24.4 (0.54) | - | 23.3 (2.8) | 31.2 (0.1) | - | - | - |
| | CS | 68.5 (8.3) | 31.8 (1.2) | **54.2 (0.8)** | **31.2 (0.8)** | **29.9 (0.3)** | **27.5 (0.2)** | 56.0 (3.7) | 0.46 (0.07) |
| FT | MMD | 67.9 (8.0) | 31.9 (1.2) | **54.2 (0.8)** | **31.2 (0.8)** | **29.9 (0.3)** | **27.5 (0.2)** | 55.9 (3.7) | **0.50 (0.06)** |
| | PRO | 66.7 (8.9) | 31.8 (1.0) | **54.2 (0.8)** | 30.4 (1.0) | 29.8 (0.3) | 27.4 (0.2) | 56.5 (2.9) | 0.49 (0.07) |
| | CS | 66.7 (8.9) | 34.0 (1.9) | 53.9 (0.7) | 27.8 (2.1) | 29.7 (0.4) | 26.9 (0.1) | 55.9 (0.8) | 0.46 (0.07) |
| E2E | MMD | 65.7 (6.7) | **34.4 (0.7)** | **54.2 (0.8)** | 27.4 (1.6) | 29.6 (0.2) | 27.0 (0.5) | 55.8 (0.7) | **0.50 (0.06)** |
| | PRO | **72.3 (9.5)** | 32.9 (0.8) | 53.8 (0.8) | 30.1 (1.2) | 29.0 (0.2) | 26.6 (0.3) | **56.4 (2.1)** | 0.49 (0.07) |

the remaining five databases for training. The performance was measured according to the original PhysioNet challenge score. This generalized intersection-over-union score assigns partial credit to misdiagnoses that result in similar treatments or outcomes. The score is then adjusted for a solution that always selects the normal/majority class and normalized for the perfect solution. Therefore, the score can have negative values and a best possible score of 1.

Table 4 reports results for the ECG experiments (see Appendix C.3 for details). For this clinical time-series data, we observe an improvement in mean score and a reduction in standard deviation over the ERM and ERM ensemble baselines across all DG tasks. We attribute poorer performance for the *PTB* dataset to the fact that it contains considerably longer recordings than other datasets (except for *INCART* which, however, contains only 75 samples) and a higher sampling rate (1000Hz vs. 500Hz and 257Hz). The negative challenge score for the *PTB-XL* dataset is due to the presence of previously unobserved labels in other datasets as well as a considerably smaller amount of data for training since the *PTB-XL* dataset comprises the majority of all samples (21,837 out of 43,101).

Table 4: Results ECG experiment. All experiments were repeated five times and the mean (standard deviation) challenge metric is reported. Higher is better. Best overall results are highlighted in **bold**.

| | | CPSC | CPSC-EXTRA | INCART | PTB | PTB-XL | G12EC |
|---|---|---|---|---|---|---|---|
| ERM | Single | 0.0840 (0.0220) | 0.2715 (0.0270) | 0.2290 (0.0059) | -8.8206 (0.3908) | -0.3373 (0.0403) | 0.2011 (0.0015) |
| | Ensemble | 0.1699 (0.0346) | 0.2488 (0.0079) | 0.2456 (0.0109) | -8.9115 (0.1023) | -0.4136 (0.0780) | 0.2079 (0.0161) |
| | CS | 0.1830 (0.0061) | 0.2950 (0.0035) | 0.1595 (0.0313) | -8.8802 (0.1069) | -0.1932 (0.0168) | 0.1853 (0.0036) |
| FT | MMD | 0.1877 (0.0077) | 0.3011 (0.0035) | 0.2100 (0.0413) | -8.8082 (0.1458) | -0.1567 (0.0211) | 0.1919 (0.0036) |
| | PROJECTION | **0.1941 (0.0050)** | **0.3135 (0.0015)** | -0.1041 (0.0015) | -8.8817 (0.0478) | -0.2166 (0.0191) | 0.2409 (0.0042) |
| | CS | 0.1067 (0.0170) | 0.2866 (0.0146) | **0.2539 (0.0289)** | -9.2947 (0.3004) | -0.1651 (0.0494) | 0.1927 (0.0080) |
| E2E | MMD | 0.1034 (0.0143) | 0.2834 (0.0228) | 0.2398 (0.0257) | -9.0600 (0.3100) | -0.1433 (0.0293) | 0.1925 (0.0067) |
| | PROJECTION | 0.1411 (0.0269) | 0.2962 (0.0065) | -0.1467 (0.0513) | **-8.5904 (0.3310)** | **-0.0178 (0.0291)** | **0.2947 (0.0117)** |

## 6 CONCLUSION AND DISCUSSIONS

We introduced the I.E.D. assumption, postulating that real-world distributions are composed of elementary distributions that remain invariant across different domains and showed that it implies an invariant structure in the solution space that enables knowledge transfer to unseen domains. Empirical results based on real-world data support the practicality of the I.E.D. assumption and that we can learn such a representation. Further, we presented a modular neural network layer consisting of Gated

Domain Units (GDUs) that leverage the I.E.D. assumption. Our GDUs can substantially improve the downstream performance of learning machines in real-world DG tasks. Across our experiments, we observed that for some datasets FT is better than E2E and vice versa. In E2E training, the feature extractor (encoder) is jointly trained with GDUs. Hence, the latent representation is stochastic during training, meaning that we have variability in the representation fed into GDUs between epochs. In contrast, in FT, the feature extractor is pretrained and always produces the same embedding. Especially with large feature extractors such as ResNet-50, learning the elementary domains can be more effective when we avoid any stochasticity in the latent representation.

**Limitations.** A major limitation of our I.E.D. assumption is to provide theoretical evidence that this assumption holds in practice. We aim to expand the scope of the theoretical understanding of the I.E.D. assumption and the GDUs. In addition, the particular theoretical setting of Albuquerque et al. (2019) (i.e., each elementary domain represents a source domain) seems promising to extend their generalization guarantee to cases where our I.E.D. assumption holds. Second, our GDU layer induces additional computational overhead due to the regularization and model size that increases as a function of the number of elementary domains. Noteworthy, our improvement is achieved with a relatively small number of elementary domains indicating that the increased complexity is not a coercive consequence of applying the DG layer. Also, the results achieved are not a consequence of increased complexity, as the ensemble baseline shows.

**Future work** We expect the I.E.D. assumption and GDUs to be adapted, yielding novel applications that tackle DG. For example, we suggest dynamically increasing the number of elementary domains during learning until their distributional variance reaches a plateau as a measure of their heterogeneity. Hence, one would learn the number of elementary domains instead of fixing the number of elementary domains prior to training.

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

# Appendices

## Table of Contents

# A    PROOFS

## A.1    PROOF OF LEMMA 1

*Proof.* The result holds trivially for $K = 1$. For $K \geq 2$ and by the I.E.D assumption, $\mathbb{P}^s(X, Y) = \sum_{j=1}^K \alpha_j \mathbb{P}_j(X, Y)$ for some $\boldsymbol{\alpha} \in \Delta^K$. Then, we can write the risk functional for each $f \in \mathcal{F}$ as $R(f) = \int \mathcal{L}(y, f(x)) d\mathbb{P}^s(x, y) = \int \mathcal{L}(y, f(x)) d(\sum_{j=1}^K \alpha_j \mathbb{P}_j(x, y)) = \sum_{j=1}^K \alpha_j \int \mathcal{L}(y, f(x)) d\mathbb{P}_j(x, y) = \sum_{j=1}^K \alpha_j R_j(f)$ where $R_j : \mathcal{F} \to \mathbb{R}_+$ is the elementary risk functional associated with the elementary distribution $\mathbb{P}_j(X, Y)$. Hence, the Bayes predictors satisfy

$$f^* \in \arg\min_{f \in \mathcal{F}} R(f) = \arg\min_{f \in \mathcal{F}} \sum_{j=1}^K \alpha_j R_j(f). \tag{A.3}$$

Since the rhs of equation A.3 corresponds to the linear scalarization of a multi-objective function $(R_1, \ldots, R_K)$, its solution (i.e., a stationary point) is Pareto-optimal with respect to these objective functions (Ma et al., 2020, Definition 3.1); see, also, (Hillermeier, 2001a;b). That is, the Bayes predictors for the data distribution that satisfies the I.E.D assumption must belong to the Pareto set $\mathcal{F}_{\text{Pareto}} := \{f^* : f^* = \arg\min_{f \in \mathcal{F}} \sum_{j=1}^K \alpha_j R_j(f), \boldsymbol{\alpha} \in \Delta^K\} \subset \mathcal{F}$.    $\square$

## A.2    PROOF OF PROPOSITION 3.1

*Proof.* Suppose we have a representation,

$$\mu_{\mathbb{P}} = \sum_{j=1}^M \beta_j \mu_{V_j} \qquad \langle \mu_{V_i}, \mu_{V_i} \rangle_{\mathcal{H}} = 0 \, \forall i \neq j, \tag{A.1}$$

i.e. $\{\mu_{V_1}, \ldots, \mu_{V_m}\}$ are pairwise orthogonal. We want to minimize the MMD by minimizing

$$\left\| \mu_{\mathbb{P}} - \sum_{j=1}^M \beta_j \mu_{V_j} \right\|_{\mathcal{H}}^2 = \underbrace{\langle \mu_{\mathbb{P}}, \mu_{\mathbb{P}} \rangle_{\mathcal{H}}}_{\|\mu_{\mathbb{P}}\|_{\mathcal{H}}^2 =} - 2\langle \mu_{\mathbb{P}}, \sum_{j=1}^M \beta_j \mu_{V_j} \rangle_{\mathcal{H}} + \langle \sum_{i=1}^M \beta_i \mu_{V_i}, \sum_{j=1}^M \beta_j \mu_{V_j} \rangle_{\mathcal{H}} \tag{A.2}$$

$$= \|\mu_{\mathbb{P}}\|_{\mathcal{H}}^2 - 2\sum_{j=1}^M \beta_j \langle \mu_{\mathbb{P}}, \mu_{V_j} \rangle_{\mathcal{H}} + \sum_{i=1}^M \sum_{j=1}^M \beta_i \beta_j \underbrace{\langle \mu_{V_i}, \mu_{V_j} \rangle_{\mathcal{H}}}_{\delta_{ij} \langle \mu_{V_i}, \mu_{V_j} \rangle_{\mathcal{H}} =} \tag{A.3}$$

$$= \|\mu_{\mathbb{P}}\|_{\mathcal{H}}^2 - 2\sum_{j=1}^M \beta_j \langle \mu_{\mathbb{P}}, \mu_{V_j} \rangle_{\mathcal{H}} + \sum_{j=1}^M \beta_j^2 \|\mu_{V_j}\|_{\mathcal{H}}^2. \tag{A.4}$$

By defining

$$\Phi(\beta) := \left\| \mu_{\mathbb{P}} - \sum_{j=1}^M \beta_j \mu_{V_j} \right\|_{\mathcal{H}}^2, \tag{A.5}$$

we can simply find the optimal $\beta_j$ by using the partial derivative

$$\frac{\partial \Phi}{\partial \beta_j} = -2\langle \mu_{\mathbb{P}}, \mu_{V_j} \rangle_{\mathcal{H}} + 2\beta_j \|\mu_{V_j}\|_{\mathcal{H}}^2 \stackrel{!}{=} 0 \tag{A.4}$$

$$\Leftrightarrow \beta_j \|\mu_{V_j}\|_{\mathcal{H}}^2 = \langle \mu_{\mathbb{P}}, \mu_{V_j} \rangle_{\mathcal{H}} \tag{A.5}$$

$$\Leftrightarrow \beta_j^* = \frac{\langle \mu_{\mathbb{P}}, \mu_{V_j} \rangle_{\mathcal{H}}}{\|\mu_{V_j}\|_{\mathcal{H}}^2}. \tag{A.6}$$

Please note that the function $\Phi$ is convex.    $\square$

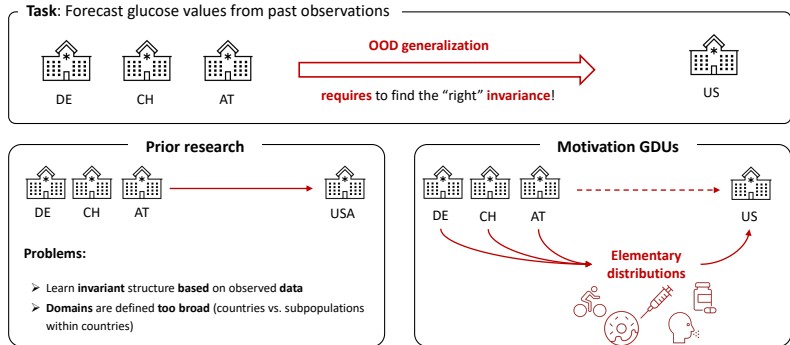

Figure 3: Visualization and motivation of our invariant elementary distribution assumption and how they can be instantiated with Gated Domain Units.

# B DETAILS ON THE GATED DOMAIN UNITS

## B.1 REAL-WORLD EXAMPLE: VISUALIZATIONS

As written in Section 2.1, we postulate that the elementary domain bases are the invariant subspaces that allow us to generalize to unseen domains. In practice, the question arises if and when elementary domains evolve. Consider that we aim to learn to predict the risk of developing Diabetes from laboratory data from Europe and then infer the risk from data from the United States of America. Naturally, factors influencing the data-generating process may change, such as the level of physical activity and nutritional habits. While, to a certain degree, these common factors remain invariant across continents, each of these factors' contributions may differ. In terms of our assumptions, we model each of these factors with a corresponding elementary distribution. Figure 3 depicts our assumption and how it differs from existing works [2].

To exploit this assumption in out-of-distribution (OOD) generalization, we developed a modular neural network layer that consists of so-called Gated Domain Units (GDUs). In Figure 4, we visualized the fundamental concept of the GDUs. Each GDU learns an embedding of an individual elementary domain that allows us to encode the domain similarities during the training. During inference, the GDUs compute similarities between observation and each of the corresponding elementary distributions, which are then used to form a weighted ensemble of learning machines. In other words, for a previously unseen individual, we aim to determine the coefficients and quantify each factor's contribution without any information about the individual's origin.

## B.2 DETAILED VIEW OF THE REGULARIZATION TERM $\Omega_D^{OLS}$

First, consider the following single term $\|\phi(\tilde{x}_i) - \sum_{j=1}^{M} \beta_{ij}\mu_{V_j}\|_{\mathcal{H}}^2$ that can be expressed as

$$\|\phi(\tilde{x}_i) - \sum_{j=1}^{M} \beta_{ij}\mu_{V_j}\|_{\mathcal{H}}^2 = \underbrace{\|\phi(\tilde{x}_i)\|_{\mathcal{H}}^2}_{(1)} - 2\underbrace{\langle\phi(\tilde{x}_i), \sum_{j=1}^{M} \beta_{ij}\mu_{V_j}\rangle_{\mathcal{H}}}_{(2)} + \underbrace{\|\sum_{j=1}^{M} \beta_{ij}\mu_{V_j}\|_{\mathcal{H}}^2}_{(3)}. \quad \text{(B.1)}$$

AD (1):

We begin with Term (1) and write $\|\phi(\tilde{x}_i)\|_{\mathcal{H}}^2$ as $\|\phi(\tilde{x}_i)\|_{\mathcal{H}}^2 = \langle\phi(\tilde{x}_i), \phi(\tilde{x}_i)\rangle_{\mathcal{H}} = k(\tilde{x}_i, \tilde{x}_i)$. We could evaluate this term using the kernel function $k$ for each data point in the batch $b$. However, since this

---

[2]Of note, Figure 3 is a complete fictive example, and we do not want to make medical implications in any way.

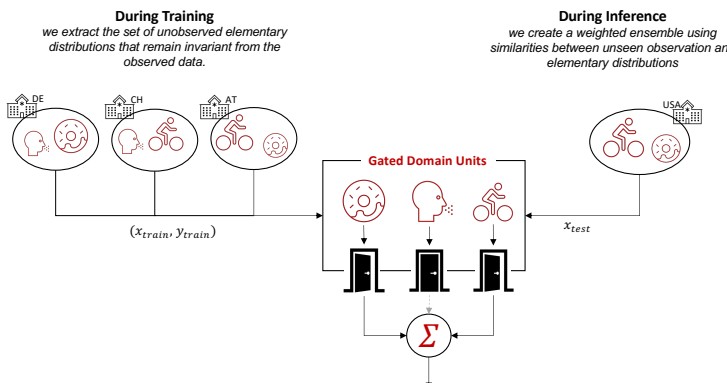

Figure 4: Visualization of the concept of the Gated Domain Unit and how they are leveraged to build distributionally weighted ensembles of learning machines.

term does not depend on the the elementary domains $\{V_1, \ldots, V_M\}$, it is unnecessary to compute this value to minimize the penalty. Thus, we obtain a similar result by minimizing the penalty without considering $\|\phi(\tilde{x}_i)\|_{\mathcal{H}}^2$ in the regularization.

AD (2):

Term (2) can be expressed as

$$\langle \phi(\tilde{x}_i), \sum_{j=1}^{M} \beta_{ij} \mu_{V_j} \rangle_{\mathcal{H}} = \sum_{j=1}^{M} \beta_{ij} \langle \phi(\tilde{x}_i), \mu_{V_j} \rangle_{\mathcal{H}} \tag{B.2}$$

Implementation-wise, the evaluation of this term requires the calculation of the inner product $\langle \phi(\tilde{x}_i), \mu_{V_j} \rangle_{\mathcal{H}}$. Since our CS and projection-based methods involve this inner product to determine the coefficients $\beta_{ij}$, we pre-compute the inner product $\langle \phi(\tilde{x}_i), \mu_{V_j} \rangle_{\mathcal{H}}$ once for a mini-batch and store these information during training to avoid multiple calculations of the same term.

Moreover, the projection-based method does not apply softmax and has a linear form. Therefore, the term (2) can be simplified even further:

$$\langle \phi(\tilde{x}_i), \sum_{j=1}^{M} \beta_{ij} \mu_{V_j} \rangle_{\mathcal{H}} = \sum_{j=1}^{M} \beta_{ij} \langle \phi(\tilde{x}_i), \mu_{V_j} \rangle_{\mathcal{H}} \tag{B.3}$$

$$= \sum_{j=1}^{M} \frac{\langle \phi(\tilde{x}_i), \mu_{V_j} \rangle_{\mathcal{H}}}{\|\mu_{V_j}\|_{\mathcal{H}}^2} \langle \phi(\tilde{x}_i), \mu_{V_j} \rangle_{\mathcal{H}} \tag{B.4}$$

$$= \sum_{j=1}^{M} \frac{\langle \phi(\tilde{x}_i), \mu_{V_j} \rangle_{\mathcal{H}}^2}{\|\mu_{V_j}\|_{\mathcal{H}}^2}. \tag{B.5}$$

AD (3):

Last, we express the term (3) as follows

$$\| \sum_{j=1}^{M} \beta_{ij} \mu_{V_j} \|_{\mathcal{H}}^2 = \sum_{j=1}^{M} \sum_{k=1}^{M} \beta_{ij} \beta_{ik} \langle \mu_{V_j}, \mu_{V_k} \rangle_{\mathcal{H}}, \tag{B.6}$$

and calculate the inner product of the domains $\langle \mu_{V_j}, \mu_{V_k} \rangle_{\mathcal{H}}$ by

$$\langle \mu_{V_j}, \mu_{V_k} \rangle_{\mathcal{H}} = \frac{1}{N^2} \sum_{l=1}^{N} \sum_{m=1}^{N} \langle \phi(v_j^l), \phi(v_k^m) \rangle_{\mathcal{H}} \tag{B.7}$$

$$= \frac{1}{N^2} \sum_{l=1}^{N} \sum_{m=1}^{N} k(v_j^l, v_k^m) =: K_{jk}, \tag{B.8}$$

where $N$ represents the number of vectors per domain basis. Note that this term does not depend on the input data $x_i$ and, hence, matrix $K_{jk}$ can be calculated once at the beginning of the optimization step and stored to be re-used for all the data point of a batch.

Combining Equation B.6 and Equation B.8 yields

$$\| \sum_{j=1}^{M} \beta_{ij} \mu_{V_j} \|_{\mathcal{H}}^2 = \sum_{j=1}^{M} \sum_{k=1}^{M} \beta_{ij} \beta_{ik} \langle \mu_{V_j}, \mu_{V_k} \rangle_{\mathcal{H}} \tag{B.9}$$

$$= \frac{1}{N^2} \sum_{j=1}^{M} \sum_{k=1}^{M} \beta_{ij} \beta_{ik} \sum_{l=1}^{N} \sum_{m=1}^{N} k(v_j^l, v_k^m) \tag{B.10}$$

$$= \sum_{j=1}^{M} \sum_{k=1}^{M} \beta_{ij} \beta_{ik} K_{jk} \tag{B.11}$$

$$= \beta_i^T K_{jk} \beta_i . \tag{B.12}$$

As a final step, we use the results for Term (1), (2), and (3) to obtain the desired regularization term

$$\Omega_D^{OLS} = \frac{1}{b} \sum_{i=1}^{b} \left( \| \phi(\tilde{x}_i) - \sum_{j=1}^{M} \beta_{ij} \mu_{V_j} \|_{\mathcal{H}}^2 \right) \tag{B.13}$$

$$= \frac{1}{b} \sum_{i=1}^{b} \left( \| \phi(\tilde{x}_i) \|_{\mathcal{H}}^2 - 2 \langle \phi(\tilde{x}_i), \sum_{j=1}^{M} \beta_{ij} \mu_{V_j} \rangle_{\mathcal{H}} + \| \sum_{j=1}^{M} \beta_{ij} \mu_{V_j} \|_{\mathcal{H}}^2 \right). \tag{B.14}$$

As mentioned above, $\| \phi(\tilde{x}_i) \|_{\mathcal{H}}^2$ is independent from the elementary domains, and, thus a constant in the regularization. Hence, we can exclude this term, which avoids additional computational effort.

### B.3 VISUALIZATION OF DG LAYER

Figure 5 depicts the layout of our DG layer.

## C EXPERIMENTS

In this section, we provide a detailed description of the DG experiment presented in Section 5. Our Digits and ECG experiments are implemented using TensorFlow 2.4.1 and TensorFlow Probability 0.12.1. For the WILDS benchmarking we use our PyTorch (version 1.11.0). All source code will be made available on GitHub https://github.com/ (TensorFlow) and https://github.com/ (PyTorch). Overall, our experiments aim to show the validity of the invariant elementary distribution (I.E.D.) assumption and the Gated Domain Units (GDUs).

For the DG layer, we considered two modes of model training: fine tuning (FT) and end-to-end training (E2E). In FT scenario, we first pre-train the FE in the ERM single fashion. Then, we extract features using the pre-trained model and pass them to the DG layer for training the latter. For the E2E training, however, the whole model including the FE and DG layer is trained jointly from the very beginning.

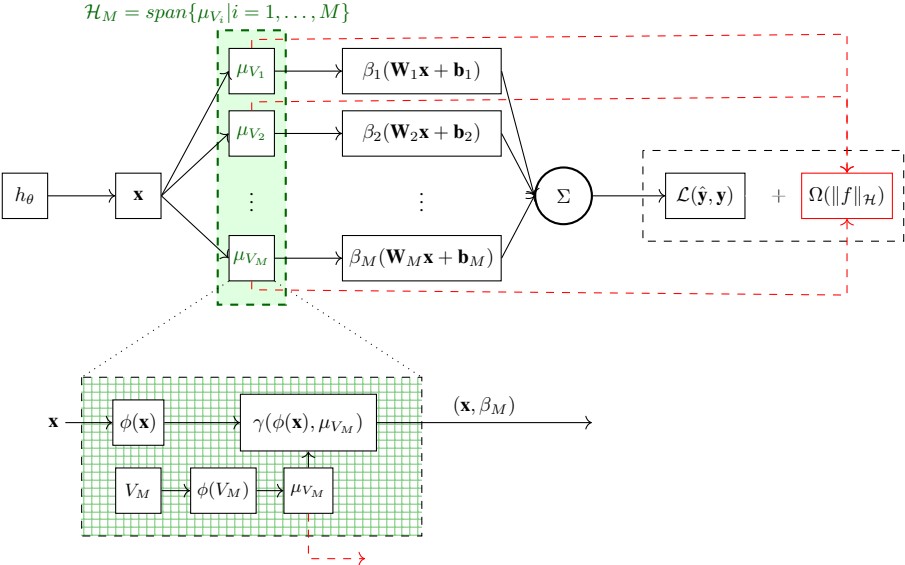

Figure 5: Domain generalization (DG) layer.

## C.1 Digits Experiment

Our experiment setup is closely related to Peng et al. (2019); Feng et al. (2020); Zhang et al. (2020); Zhao et al. (2018). Each dataset, except USPS, is split into training and test sets of 25,000 and 9,000 images, respectively. For USPS, we take the whole dataset for the experiment since it contains only 9,298 images[3]. Our experimental setup regarding datasets, data loader, and FE are based on existing work (Feng et al., 2020; Peng et al., 2019). The structure of the FE is summarized in Table 5 and the subsequent learning machine is a dense layer.

In the Empirical Risk Minimization (ERM) single experiment, we add a dense layer with 10 outputs (activation=*tanh*) as a classifier to the FE. In the Empirical Risk Minimization (ERM) ensemble experiment, we add $M$ classification heads (a dense layers with 10 outputs and *tanh* activation each) to the FE and average their output for the final prediction. This sets a baseline for our DG layer to show performance gain against the ERM model with the same number of learning machines.

For training, we resorted to the Adam optimizer with a learning rate of 0.001. We used early stopping and selected the best model weights according to the validation accuracy. For the validation data, we used the combined test splits only of the respective source datasets. The batch size was set to 512. Although the DG layer requires more computation resources than the ERM models, all digits experiments were conducted on a single GPU (NVIDIA GeForce RTX 3090).

**Heuristics for main parameter of DG layer**    From a practical perspective, our layer requires choosing two main hyper-parameters: the number of elementary domains $M$ and since we use the characteristics Gaussian kernel the corresponding parameter $\sigma$. The parameter $M$ determines the size of the ensemble of learning machines and, thus, for deep learning models, their overall network size. As a heuristic to choose $M$, we suggest to cluster the output of a pre-trained FE. In the following, we provide an example. We pre-trained the FE for the test domain *MNIST-M* and pass the source data through this FE, which we cluster with the k-means algorithm. Subsequently, we analyse three different metrics (Calinski Harabasz score, Davies Bouldinn score, and Silhouette score) to select the optimal number of clusters as the basis to choose $M$. All scores yielded an accordance between four to five clusters. Therefore, we set $M$ to five and observed in Table 2 in Section 5 strong results in the generalizing to the unseen test domain *MNIST-M*.

---

[3]We used the digits data from https://github.com/FengHZ/KD3A [last accessed on 2022-05-17, available under MIT License.] published in Feng et al. (2020).

Table 5: Feature Extractor used for the Digits Experiment

| FEATURE EXTRACTOR | |
|---|---|
| LAYER TYPE | OUTPUT SHAPE |
| 2D-CONVOLUTIONAL LAYER | (32, 32, 64) |
| BATCH NORMALIZATION | (32, 32, 64) |
| MAXPOOLING 2D | (16, 16, 64) |
| 2D-CONVOLUTIONAL LAYER | (16, 16, 64) |
| BATCH NORMALIZATION | (16, 16, 64) |
| MAXPOOLING 2D | (8, 8, 64) |
| 2D-CONVOLUTIONAL LAYER | (8, 8, 128) |
| BATCH NORMALIZATION | (8, 8, 128) |
| MAXPOOLING 2D | (4, 4, 128) |
| FLATTEN | (2048) |
| DENSE LAYER | (3072) |
| BATCH NORMALIZATION | (3072) |
| DROPOUT | (3072) |
| BATCH NORMALIZATION | (2048) |
| DENSE LAYER | (2048) |

As for the parameter $\sigma$, we resort to the median heuristic proposed in (Muandet et al., 2016) that is $\sigma^2 = \text{median}\{ \parallel \tilde{x}_i - \tilde{x}_j \parallel^2 : i, j = 1, \ldots, n\}$. While both heuristics require a pre-trained FE, cross-validation can act as a reasonable alternative. The hyper-parameters relevant for the DG layer are summarized in Table 6. In the FT setting, we applied the median heuristics presented above to estimate $\sigma$ of the Gaussian kernel function, where the estimator is denoted as $\hat{\sigma}$. Since median heuristic is not applicable for the E2E scenario, $\sigma$ was fixed to 7.5 for E2E.

Note that our approach to choose the relevant parameters was kept very general to show the feasibility of the I.E.D. assumption and the generalization ability of GDUs and, most importantly, to provide easy-to-reproduce results. During training, additional epoch metrics can be subscribed using our custom *DG layer callback*, which may help to choose the model parameters. Furthermore, we observed that the elementary domains become naturally orthogonal during the experiments, and thus, we set $\lambda_{ORTH}$ relatively small. Since the orthogonal regularization puts additional computational burden, one could omit this term completely to speed up training.

Table 6: Parameters for DG Layer in Digits and Digit-DG Experiments for the Fine Tuning (FT) and End-to-end training (E2E) Settings. In case of Projection, we chose the spectral restricted isometry property (SRIP) as the orthogonal regularization $\Omega_D^\perp$.

| | EXPERIMENT | M | N | $\lambda_{L_1}$ | $\lambda_{OLS}$ | $\lambda_{ORTH}$ | $\sigma$ | $\kappa$ |
|---|---|---|---|---|---|---|---|---|
| FT | CS | 5 | 10 | $1e^{-3}$ | $1e^{-3}$ | - | $\hat{\sigma}$ | 2 |
| | MMD | 5 | 10 | $1e^{-3}$ | $1e^{-3}$ | - | $\hat{\sigma}$ | 2 |
| | PROJECTION | 5 | 10 | $1e^{-3}$ | $1e^{-3}$ | $1e^{-8}$ | $\hat{\sigma}$ | - |
| E2E | CS | 5 | 10 | $1e^{-3}$ | $1e^{-3}$ | - | 7.5 | 2 |
| | MMD | 5 | 10 | $1e^{-3}$ | $1e^{-3}$ | - | 7.5 | 2 |
| | PROJECTION | 5 | 10 | $1e^{-3}$ | $1e^{-3}$ | $1e^{-8}$ | 7.5 | - |

**Digit-DG Benchmark**    In previous research, the aforementioned digits data is not only used for domain adaptation (DA), but also for domain generalization (DG) methods. For the latter, Zhou et al. (2021b) and Li et al. (2021) introduced Digit-DG dataset and the evaluation protocol to benchmark seven DG methods and ERM [4]. Unlike the Digits experiment described above, Digit-DG dataset from Zhou et al. (2021b) and Li et al. (2021) consists of only four datasets (without USPS) and a different FE summarized in Table 8. Therefore, we follow their instructions to conduct a fair comparison and ensure reproducibility. For the hyper-parameters, however, we kept the same values that we used for the Digits experiment, see Table 6.

---

[4]Results were reported by Zhou et al. (2021b) and Li et al. (2021). Of note, both authors did not report the standard deviation on their results.

Table 7: Results Digits experiment. All experiments were repeated ten times and the mean (standard deviation) accuracy is reported. Best results according to the mean accuracy are highlighted in **bold**.

| | | MNIST | MNIST-M | SVHN | USPS | SYN |
|---|---|---|---|---|---|---|
| *ERM* | *Single* | *97.98 (0.34)* | *63.00 (3.20)* | *70.18 (2.74)* | *93.70 (1.74)* | *83.62 (1.47)* |
| | *Ensemble* | *98.21 (0.39)* | *62.87 (1.50)* | *72.01 (3.59)* | *95.16 (0.89)* | *83.80 (1.22)* |
| **FT** | CS | 98.53 (0.16) | 68.55 (0.80) | 78.90 (1.41) | 95.83 (0.50) | 88.39 (0.82) |
| | MMD | 98.60 (0.08) | 68.62 (0.70) | 79.20 (2.01) | **96.24 (0.71)** | 88.27 (0.41) |
| | Projection | 98.57 (0.17) | 68.56 (0.91) | 79.34 (0.72) | 96.24 (0.71) | 88.58 (0.53) |
| **E2E** | CS | 98.62 (0.19) | **69.25 (0.61)** | **79.42 (1.27)** | 96.17 (0.52) | 87.92 (0.84) |
| | MMD | 98.58 (0.16) | 69.04 (0.83) | 79.20 (0.90) | 96.00 (0.44) | 88.18 (0.86) |
| | Projection | **98.67 (0.12)** | 68.67 (0.98) | 78.56 (1.68) | 96.24 (0.77) | **88.77 (0.48)** |

Table 8: Feature Extractor used for the Digit-DG Benchmark Experiment

| FEATURE EXTRACTOR | |
|---|---|
| LAYER TYPE | OUTPUT SHAPE |
| 2D-CONVOLUTIONAL LAYER | (32, 32, 64) |
| MAXPOOLING 2D | (16, 16, 64) |
| 2D-CONVOLUTIONAL LAYER | (16, 16, 64) |
| MAXPOOLING 2D | (8, 8, 64) |
| 2D-CONVOLUTIONAL LAYER | (8, 8, 64) |
| MAXPOOLING 2D | (4, 4, 128) |
| 2D-CONVOLUTIONAL LAYER | (8, 8, 64) |
| MAXPOOLING 2D | (4, 4, 128) |
| 2D-CONVOLUTIONAL LAYER | (4, 4, 64) |
| MAXPOOLING 2D | (2, 2, 64) |
| FLATTEN | (256) |

As a first method, we consider the CCSA (Classification and Contrastive Semantic Alignment) method, which learns a domain-invariant representation by utilizing the CCSA loss (Motiian et al., 2017). Second, MMD-AAE (Maximum Mean Discrepancy-based Adverserial Autoencoders) extends adverserial autoencoders by a maximum mean discrepancy regularization to learn a domain-invariant feature representation (Li et al., 2018b). CrossGrad (Cross-Gradient) augments data by perturbating the input space using the cross-gradients of a label and domain predictor (Shankar et al., 2018). Another augmentation-based DG method is L2A-OT (Learning to Augment by Optimal Transport) (Zhou et al., 2021b). Specifically, a data generator trained to maximize the optimal transport distance between source and pseudo domains, is used to augment the source data. All aforementioned methods rely on the availability of domain information such as domain labels. To benchmark our layer to a method for DG without domain information, we resort to the JiGen (Jigsaw puzzle based Generalization) method (Carlucci et al., 2019). JiGen introduces an auxiliary loss for solving jigsaw task during training. Further, we use the adaptive and non-adaptive stochastic feature augmentation (SFA-S and SFA-A, respectively) method proposed by Li et al. (2021). In principle, both method augment the latent feature embedding of a FE using random noise.

Our results are summarized in Table 9. As noted by Li et al. (2021), it is challenging to outperform augmentation-based DG methods. In addition, SFA-A and SF-S are computationally light (i.e., only adding random noise to the feature embedding) and do not require domain information (Li et al., 2021). Nevertheless, our layer achieves competitive results even against the strongest baselines in all DG tasks without requiring domain information.

Table 9: Results of the Digits-DG experiment. All experiments were repeated ten times. Methods are classified into augmentation-based (A) and non-augmentation-based (B) as well as DG with (✓) and without (✗) domain information according to Li et al. (2021). Best results according to the mean accuracy are highlighted in **bold**.

| | | MNIST | MNIST-M | SVHN | SYN | CATEGORIE | DOMAIN INFORMATION |
|---|---|---|---|---|---|---|---|
| | | | *Results reported by Li et al. (2021)* | | | | . |
| *ERM* | | *95.8* | *58.8* | *61.7* | *78.6* | B | ✗ |
| *CCSA* (MOTIIAN ET AL., 2017) | | *95.2* | *58.2* | *65.5* | *79.1* | B | ✓ |
| *MMD-AAE* (LI ET AL., 2018B) | | *96.5* | *58.4* | *65.0* | *78.4* | B | ✓ |
| *CrossGrad* (SHANKAR ET AL., 2018) | | *96.7* | *61.1* | *65.3* | *80.2* | A | ✓ |
| *L2A-OT* (ZHOU ET AL., 2021B) | | *96.7* | *63.9* | *68.6* | *83.2* | A | ✓ |
| *SFA-S* (LI ET AL., 2021) | | *96.7* | *66.3* | *68.8* | *85.1* | A | ✗ |
| *SFA-A* (LI ET AL., 2021) | | *96.5* | *66.5* | *70.3* | *85.0* | A | ✗ |
| *JiGen* (CARLUCCI ET AL., 2019) | | *96.5* | *61.4* | *63.7* | *74.0* | B | ✗ |
| | | | *Gated Domain Units (GDUs)* | | | | . |
| | CS | 97.5 | 68.9 | 74.0 | 85.5 | B | ✗ |
| **FT** | MMD | 97.6 | 69.0 | 74.2 | 84.3 | B | ✗ |
| | PROJECTION | 97.7 | 69.1 | 74.1 | 86.1 | B | ✗ |
| | CS | 97.6 | 69.4 | **75.9** | **86.5** | B | ✗ |
| **E2E** | MMD | 97.6 | **69.5** | 75.6 | **86.5** | B | ✗ |
| | PROJECTION | **97.8** | 66.7 | 73.4 | 84.0 | B | ✗ |

## C.2 ABLATION STUDY

### C.2.1 MAIN COMPONENTS OF THE GATED DOMAIN UNIT

We chose the Digits dataset to conduct an ablation study, which is organized as follows: (1) ablation of the regularization terms presented in Section 3, (2) effect of the orthogonal regularization for projection-based generalization, and (3) affect on the FE's output.

As a reminder, we introduced the regularization to be dependent on the form of generalization (i.e., domain similarity measures or projection-based generalization in Section 3). For the domain similarity measure case, the regularization is

$$\Omega_D\big(\|g\|_{\mathcal{H}}\big) = \lambda_{OLS}\Omega_D^{OLS}\big(\|g\|_{\mathcal{H}}\big) + \lambda_{L_1}\Omega_D^{L_1}(\|\gamma\|), \quad\quad\text{(C.1)}$$

where $\lambda_{OLS}, \lambda_{L_1} \geq 0$. In the case of projection, the regularization is given by

$$\Omega_D\big(\|g\|_{\mathcal{H}}\big) = \lambda_{OLS}\Omega_D^{OLS}\big(\|g\|_{\mathcal{H}}\big) + \lambda_{ORTH}\Omega_D^{\perp}\big(\|g\|_{\mathcal{H}}\big) \quad\quad\text{(C.2)}$$

with $\lambda_{OLS}, \lambda_{ORTH} \geq 0$. Although one can additionally choose the sparse regularization in projection-based generalization, we set the focus in the ablation study on the two main regularization terms that are the OLS and orthogonal regularization. For (1) we vary in Equation C.1 and Equation C.2 the corresponding weights $\lambda_1$ and $\lambda_2$ in the interval of $[0; 0.1]$ and display the mean classification accuracy for the most challenging classification task of MNSIT-M in the form of a heatmap. In Figures 6-8, we see that the classification accuracy remains on an overall similar level which indicates that the DG layer is not very sensitive to the hyper-parameter change for MNIST-M as the test domain. Nevertheless, we observe that ablating the regularization terms by setting the corresponding weights to zero decreases the classification results and the peaks in performance occur when the regularization is included during training of the DG layer.

Applying the DG layer comes with additional overhead, especially the regularization that ensures the orthogonality of the elementary domain bases. This additional effort raises a question whether ensuring the theoretical assumptions outweigh the much higher computational effort. Thus, in a second step, we analyze how the orthogonal regularization affects the orthogonality of the elementary domain bases (i.e., spectral restricted isometry property (SRIP) value) and the loss function (i.e., categorical cross-entropy).

In Figure 10, we depict the mean and standard deviation of the SRIP value and loss over five runs for 40 epochs. The SRIP value can be tracked during training with the DG layer's callback functionalities. First, we observe that the elementary domains are almost orthogonal when initialized. Training the layer leads in the first epochs to a decrease in orthogonality. This initial decrease happens because

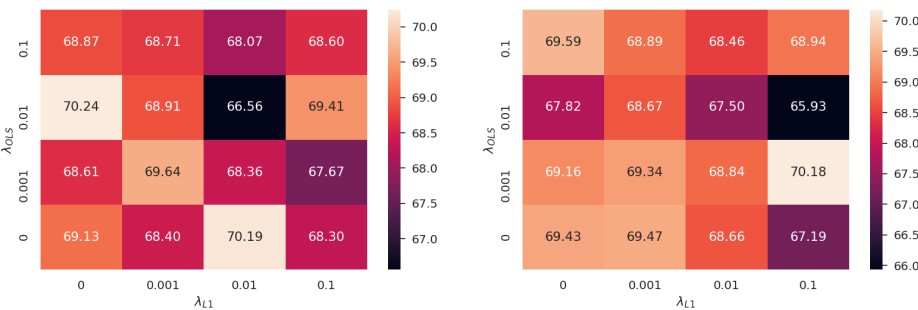

Figure 6: Classification results for varying $\lambda_{L_1}$ and $\lambda_{OLS}$ in the interval of $[0; 0.1]$ for FT (left) and E2E (right) CS on MNIST-M.

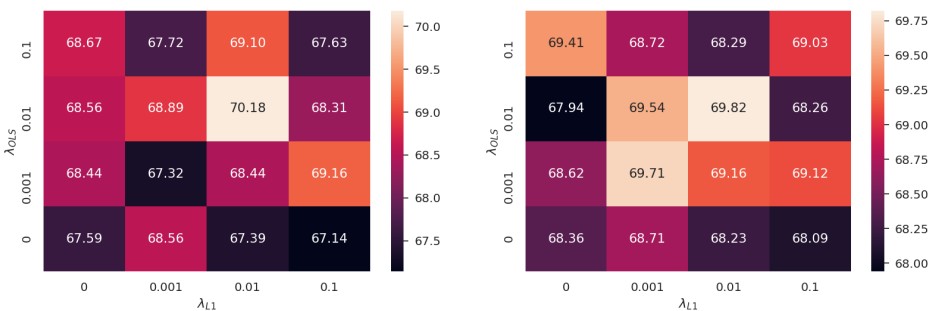

Figure 7: Classification results for varying $\lambda_{L_1}$ and $\lambda_{OLS}$ in the interval of $[0; 0.1]$ for FT (left) and E2E (right) MMD on MNIST-M.

cross-entropy has a stronger influence on the optimization than regularization in the first epochs. After five epochs, the cross-entropy decrease to a threshold when the regularization becomes more effective and the orthogonality of the elementary domain bases increases again. In Figure 10, we also observe that ablating the orthogonal regularization, while leading to better orthogonality of the domains, does not significantly affect the overall cross-entropy during training.

Finally, we project the output of the FE trained with a dense layer (ERM) and with the DG layer by t-SNE (t-distributed stochastic neighbor embedding) in Figure 11. The GDU-trained FE yields more concentrated and bounded clusters in comparison to the one trained by ERM. Hence, we observe a positive effect on the representation learned by the FE.

### C.2.2 INTERPRETATION OF THE ELEMENTARY DOMAINS

We analyze the learned elementary domains in the digits experiment based on two visualizations, and choose the maximum mean discrepancy (MMD) as the similarity measure and *MNIST-M* as the test domain. The first visualization depicts the MMD between the datasets (i.e., *MNIST*, *MNIST-M*, *SVHN*, *USPS*, and *Synthetic Digits (SYN)*) and the learned elementary domains (i.e., $V_1 - V_5$) as a heatmap (see Figure 12 (left)). The heatmap indicates that the source and test domains are close to one another in terms of the MMD. Hence, we expect that their closeness reflects in the learning of the elementary domains. In other words, we expect that each elementary domains contributes similarly to the source and test domains (i.e., the coefficients $\beta$ are similar for each of these domains). In Section 3.1, we derive the coefficients by applying a kernel softmax function to the negative MMD distances. Since the MMD distances between the source / test domains and the elementary domains are similar, the coefficients will be similar too. We conclude that the learned elementary domains represent the same distributional characteristics that existed among the source and test domains.

In the second visualisation, we show the t-SNE (t-distributed stochastic neighbor embedding) of the feature extractor output for each source and test domain alongside the elementary domains in Figure 12 (right). First, we observe that the learned elementary domain bases form distinctive clusters. We

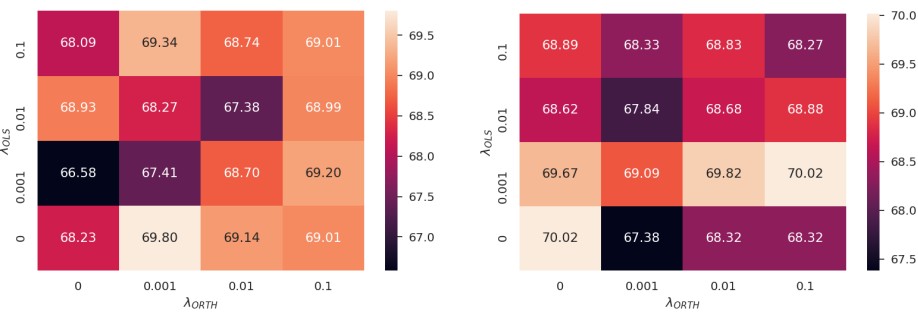

Figure 8: Classification results for varying $\lambda_{ORTH}$ and $\lambda_{OLS}$ in the interval of $[0; 0.1]$ for FT (left) and E2E (right) Projection on MNIST-M.

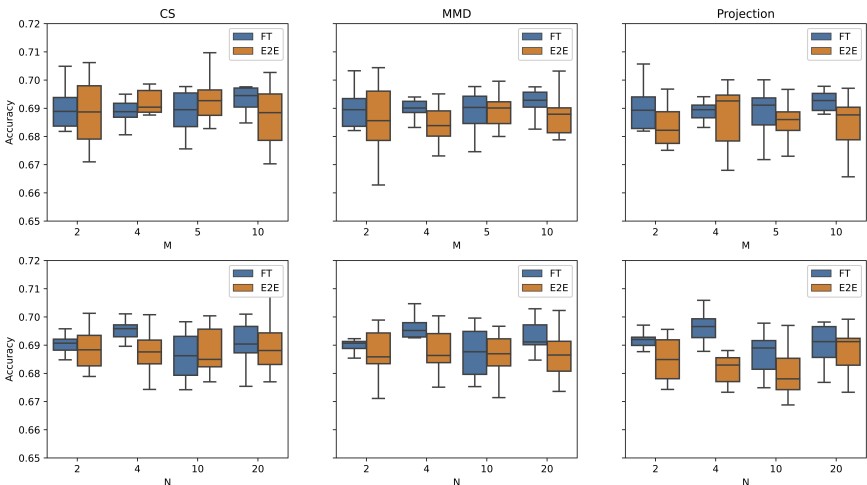

Figure 9: Mean and standard deviation of classification accuracy over 10 runs for varying number of elementary domains ($M$, upper panel) and varying number of vector for each domain basis ($N$, lower panel) for MNIST-M dataset.

see these clusters as a validation of our hypothesis that each GDU learns to mimic samples generated from a corresponding elementary distribution as pointed out in Section 2.2. However, we can not answer whether and where these elementary distributions occur in the real world. Moreover, these elementary distributions yet lack interpretability.

In summary, the MMD heatmap and t-SNE embeddings of the learned elementary and source domains on Figure 12 indicate that the GDUs learn to represent distributional structures in the dataset.

### C.3 ECG EXPERIMENT

We adopted the task of multi-label binary classification of 12-lead electrocardiogram (ECG) signals combined from 6 different sources introduced in the *PhysioNet/Computing in Cardiology Challenge 2020*[5] (Perez Alday et al., 2021; Goldberger et al., 2000; Perez Alday et al., 2020). Each ECG recordings is annotated with 24 binary labels indicating whether or not a certain cardiac abnormality is present. The data is aggregated from 6 different databases and contains 43,101 recordings sampled

---

[5] https://physionetchallenges.org/2020/ [last accessed on 2021-03-10, available under Creative Commons Attribution 4.0 International Public License].

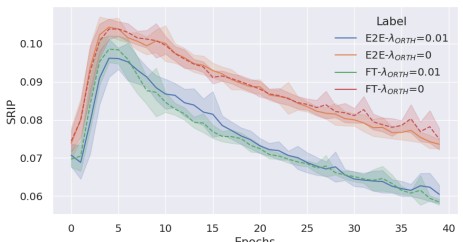 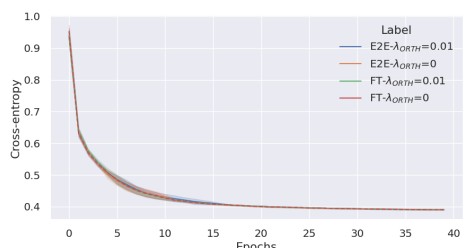

Figure 10: Effect of omitting the orthogonal regularization term $\Omega_D^{\perp}$. Spectral restricted isometry property (SRIP) (left) and categorical cross-entropy (right) with and without orthogonal regularization and their evolution during training for MNSIT-M dataset. The mean and standard deviation presented for End-to-end (E2E) and Fine-tuning (FT) training scenarios are calculated over 10 runs.

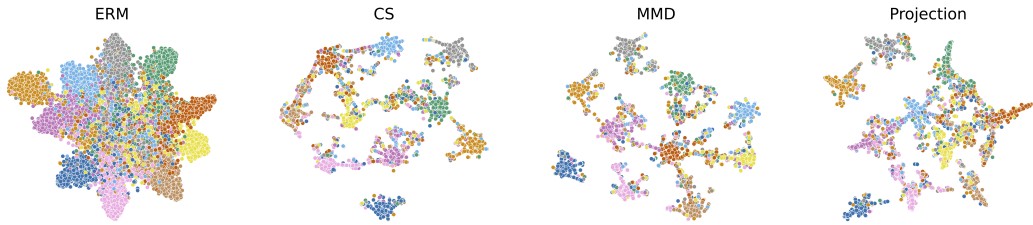

Figure 11: Visualization of t-SNE Embedding on unseen Synthetic Digits Dataset. Colors encode true label.

with various sampling frequencies, number of subjects, and lengths. Table 10 summarizes most important details about the data sources for this experiment.

According to the original challenge score, we measure the performance in terms of the generalized Intersection-over-Union (IoU) score where partial credit is assigned to misdiagnoses that result in similar treatments or outcomes. The score is defined as

$$score := \frac{y^T \cdot W \cdot \hat{y}}{y \cup \hat{y}}, \tag{C.3}$$

where $y, \hat{y} \in \{0, 1\}^{24}$ represent actual labels and predicted labels and $W$ stands for the partial credit-assignment matrix provided as a part of the challenge description. Note that in case of identity matrix $W$ the score is exactly the Intersection-over-Union (IoU) score. The score is then adjusted for a solution $y_{majority}$, which always predicts the normal/majority class, and is moreover normalized

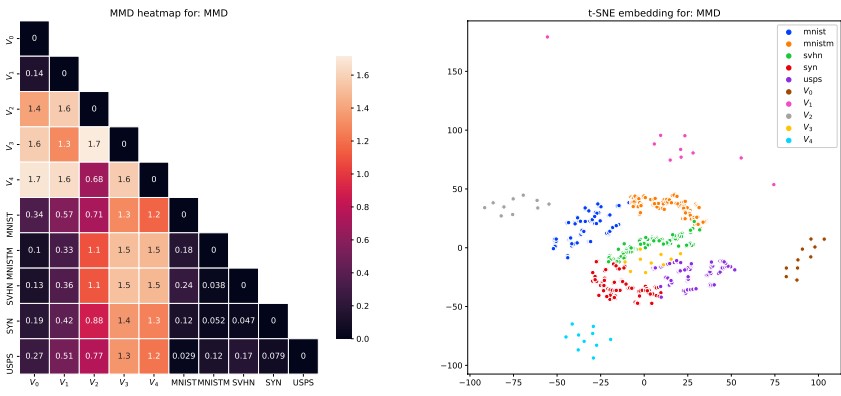

Figure 12: MMD heatmap (left) and t-SNE embedding (right) for the test domain MNIST-M.

Table 10: ECG Data Sources Details.

| DATASET | NUMBER OF SAMPLES | ECG LENGTH [SEC] | FREQUENCY | COUNTRY |
|---|---|---|---|---|
| CPSC | 6,877 | 6 TO 60 | 500 Hz | CHINA |
| CPSC-EXTRA | 3,453 | 6 TO 60 | 500 Hz | CHINA |
| INCART | 75 | 1,800 | 257 Hz | RUSSIA |
| PTB | 549 | UP TO 120 | 1,000 Hz | GERMANY |
| PTB-XL | 21,837 | 10 | 500 Hz | GERMANY |
| G12EC | 10,344 | 10 | 500 Hz | USA |

for the perfect solution $y$. Therefore, the final score can have negative values and the best possible score of 1 and is formalized as

$$adjusted \ \ score := \frac{score(y, \hat{y}) - score(y, y_{majority})}{score(y, y) - score(y, y_{majority})}. \tag{C.4}$$

As a pre-processing step, we down-sampled all the signals to 125 Hz and applied Z-score, random amplification and random stretching according to Vicar et al. (2020). For that we partially adopted the code provided by the authors[6]. Additionally, we cropped each signal to its first 15,000 points if the signal was too long (mostly applied to *INCART* database). Each dataset was randomly split into train and validation parts with 3:1 ratio. During each experiment, we used the train splits of 5 databases for training and utilized the validation splits of the training databases for early stopping. The hold-out 6-th database was used for inference and testing only.

Table 11 describes the architecture of FE used for the task. Since the provided ECG recordings have different lengths, we used TensorFlow padded batching, which is padding all the recordings in a batch to the length of the longest sequence in the batch. Therefore, input from different batches can have different lengths so the spatial dimensions of the 1D-Convolutional layers are not predefined and are presented as *.

Table 11: Feature Extraction Architecture used for the ECG Experiment is an adapted Version of LeNet Architecture for 1D input Signals. Note that ECG recordings have variable lengths, therefore, the spatial dimension is not defined and denoted as *.

| FEATURE EXTRACTOR | |
|---|---|
| LAYER TYPE | OUTPUT SHAPE |
| 1D-CONVOLUTIONAL LAYER (KERNEL SIZE=*3*, ACTIVATION=*relu*) | (*, 32) |
| BATCH NORMALIZATION | (*, 32) |
| MAXPOOLING 1D (POOL SIZE=*2*, STRIDES=*2*) | (*, 32) |
| 1D-CONVOLUTIONAL LAYER (KERNEL SIZE=*2*, ACTIVATION=*relu*) | (*, 64) |
| BATCH NORMALIZATION | (*, 64) |
| GLOBAL AVERAGE POOLING 1D | (64) |
| DENSE LAYER (ACTIVATION=*relu*) | (100) |
| DENSE LAYER (ACTIVATION=*relu*) | (100) |

We used the Adam optimizer to optimize weighted binary cross-entropy loss defined as $-(w_{pos} \cdot y \cdot \log \hat{y}) + (1 - y) \cdot \log (1 - \hat{y})$. Positive weights $w_{pos}$ are defined per class based on the training split data inversely proportional to the frequency of positive labels for each class. A learning rate was initially set to 0.001 and during the training reduced by the factor of 0.2 if the training loss was not improving for 10 epochs. We also applied early stopping and restored model weights to the best model according to the validation accuracy after the training end. Since each input samples for this experiment have a larger size than the previous one, we decreased the batch size to 64. Each ECG experiment was performed on a single GPU (Nvidia GTX 1080 Ti). The parameters relevant for the DG layer are summarized in Table 12. We have to emphasize that we did not perform extensive hyper-parameter tuning since our goal was to show the feasibility of the I.E.D. assumption and GDUs while keeping the experiments reproducible.

---

[6] https://github.com/tomasvicar/BUTTeam [last accessed on 2022-05-17, available under BSD 2-Clause License].

Table 12: Parameters for DG Layer in ECG experiments for the Fine Tuning (FT) and End-to-end training (E2E) Settings.

| | Experiment | M | N | $\lambda_{L_1}$ | $\lambda_{OLS}$ | $\lambda_{ORTH}$ | $\sigma$ | $\kappa$ |
|---|---|---|---|---|---|---|---|---|
| | CS | 10 | 10 | $1e^{-3}$ | $1e^{-3}$ | - | 5.5 | 2 |
| FT | MMD | 10 | 10 | $1e^{-3}$ | $1e^{-3}$ | - | 5.5 | 2 |
| | Projection | 10 | 10 | $1e^{-3}$ | $1e^{-3}$ | $1e^{-6}$ | 5.5 | - |
| | CS | 10 | 10 | $1e^{-3}$ | $1e^{-3}$ | - | 5.5 | 2 |
| E2E | MMD | 10 | 10 | $1e^{-3}$ | $1e^{-3}$ | - | 5.5 | 2 |
| | Projection | 10 | 10 | $1e^{-3}$ | $1e^{-3}$ | $1e^{-6}$ | 5.5 | - |

## C.4 WILDS Benchmarking Experiments

For comparison of our approach and benchmarking, we followed the standard procedure of WILDS experiments, described in Koh et al. (2021). As a technical note, all WILDS experiments have been implemented in Pytorch (version $>= 1.7.0$) based on the specifications made in Koh et al. (2021) and their code published on https://github.com/p-lambda/wilds [last accessed on 2022-05-17, available under MIT License]. The results for the benchmarks were retrieved from the official leaderboard https://wilds.stanford.edu/leaderboard/ [last accessed on 2022-09-26].

**Camelyon17** In medical applications, the goal is to apply models trained on a comparatively small set of hospitals to a larger number of hospitals. For this application, we study images of tissue slides under a microscope to determine whether a patient has cancer or not. Shifts in patient populations, slide staining, and image acquisition can impede model accuracy in previously unseen hospitals. Camelyon17 comprises images of tissue patches from five different hospitals. While the first three hospitals are the source domains (302,436 examples), the forth and fifth are the validation (34,904 examples) and test domain (85,054 examples), respectively.

We deviate from the specifications made in (Koh et al., 2021) in terms of the FE. We use the FE from Feng et al. (2020); Peng et al. (2019) since we observed a higher mean accuracy and faster training than with the by Koh et al. (2021) originally proposed DenseNet-121 FE (Huang et al., 2017). We trained the FE from scratch. Both, ERM and the DG were trained over 250 epochs with early stopping, a learning rate of 0.001, which is reduced by a factor of 0.2 if the cross-entropy loss has not improved after 10 epochs. All results were aggregated over ten runs.

**FMoW** Analyzing satellite images with machine learning (ML) models may enable novel possibilities in tackling global sustainability and economic challenges such as population density mapping and deforestation tracking. However, satellite imagery changes over time due to human behavior (e.g., infrastructure development), and the extent of change is different in each region. The Functional Map of the World (FMoW) dataset consists of satellite images from different continents and years: training (76,863 images; between 2002–2013), validation (19,915 images; between 2013 and 2016), and test (22,108 images, between 2016–2017). The objective is to determine one of 62 building types (e.g., shopping malls) and land-use.

As instructed in Koh et al. (2021), we used the DenseNet-121 pre-trained on ImageNet without L2-regularization. For the optimization, we use the Adam optimizer with a learning rate of 1e-4, which is decayed by a factor of 0.96 per epoch. The models were trained for 50 epochs with early stopping and a batch size of 64. Additionally, we report the worst-region accuracy, which is a specific metric used for FMoW. This worst-region accuracy reports the worst accuracy across the following regions: Asia, Europe, Africa, America, and Oceania (see Koh et al. (2021) for the details). Again, we report the results over three runs.

**Amazon** Recent research shows that consumer-facing machine learning application large performance disparities across different set of users. To study this performance disparities, WILDS (Koh et al., 2021) leverages a variant of the Amazon Review dataset. The Aamazon-WILDS dataset is composed of data from 3,920 domains (number of reviewers) and the task is a multi-class sentiment classification, where the model receives a review text and has to predict the rating from one to five.

To split this dataset, a between training, validation, and test disjoint set of reviewers is used: training (245,502 reviews from 1,252 reviewers), validation (100,050 reviews from 1,334 reviewers), test (100,050 reviews from 1,334 reviewers).

For the experiments and baseline models, we use the specifications made in Koh et al. (2021). As for the FE, we used DistilBERT-base-uncased models. For ERM, we use a batch size of 8, learning rate 1e-5, L2 regularization of 0.01, 3 epochs with early stopping and 512 as the maximum length of tokens. For training the DG layer, we used the same specifications as made for ERM. The performance is measured in 10th percentile accuracy.

**iWildsCam** Wildlife camera traps offer an excellent possibility to understand and monitor biodiversity loss. However, images from different camera traps differ in illumination, color, camera angle, background, vegetation, and relative animal frequencies. We use the iWildsCam dataset consisting of 323 different camera traps positioned in different locations worldwide. In the dataset, we refer to different locations of camera traps as different domains, in particular 243 training traps (129,809 images), 32 validation traps (14,961 images), and 48 test traps (42,791 images). The objective is to classify one of 182 animal species.

Following the instructions by Koh et al. (2021), we used again the ResNet50 pre-trained on ImagNet (He et al., 2016). For ERM, we used a learning rate of 3e-5 and no L2-regularization. The models were trained for 12 epochs with a batch size of 16 with the Adam optimizer. In addition to the accuracy, we report the macro F1-score to evaluate the performance on rare species (see Koh et al. (2021) for details). All results were aggregated over three runs.

**RxRx1** In biomedical research areas such as genomics or drug discovery, high-throughput screening techniques generate a vast amount of data in several batches. Because experimental designs cannot fully mitigate the effects of confounding variables like temperature, humidity, and measurements across batches, this creates heterogeneity in the observed datasets (commonly known as batch effect). The RxRx1 dataset comprises images obtained by fluorescent microscopy from 51 domains (disjoint experiments): training (40,612 images, 33 domains), validation (9,854 images, 4 domains), and test (34,432 images, 14 domains). The aim is to classify one of 1,139 genetic treatments. All results were aggregated over three runs.

We conducted the RxRx1 experiments in accordance with the specifications made in (Koh et al., 2021). As for the FE, we, thus, used the ResNet50 pre-trained on ImagNet (He et al., 2016). We trained the models using AdamW with default parameters $\beta_1 = 0.9$ and $\beta_2 = 0.999$ using a learning rate of 1e-4 and a L2-regularization with strength 1e-5 for 90 epochs with a batch size of 75. We scheduled the learning rate to linearly increase in the first ten epochs and then decreased it following a cosine rate. For trainingthe DG layer, we chose the same parameters as for the ERM. All results were aggregated over three runs.

**OBG-MolPCBA** In biomedical research, machine learning has the potential to accelerate drug discovery while reducing the experimental overhead due to lowering the number of experiments required. However, to leverage the potential of machine learning, the models need to generalize to molecules structurally different from those seen during training. To study this OOD generalization across molecule scaffolds, we use the OGB-MolPCBA dataset. This dataset is split into the following subsets according to the scaffold structure: training (44,930 domains), validation (31,361 domains), and test (43,739 domains). The task is to classify the presence/absence of 128 biological activities based on a graph representation of a molecule.

In line with Koh et al. (2021), we use a Graph Isomorphism Network (GIN) combined with virtual nodes as the FE. For training ERM and our DG, we use the default parameters: five GNN layers with a dimensionality of 300 and a learning rate of 0.001. We train for 100 epochs using early stopping. As for the performance, we report the mean and standard deviation of the average precision across all scaffolds (domains) over three runs.

**CivilComments** In the last decades, users have generated a vast amount of text on the Internet, some of which contain toxic comments. Machine learning has been leveraged for automatic text review to flag toxic comments. However, the models are prone to learn spurious correlations between toxicity and information on demographics in the comment, which causes the model performance to

drop in specific subpopulations. To study this OOD task, we leverage the modified CivilComment dataset from Koh et al. (2021). Based on text input, the task is to predict a binary label, toxic or non-toxic. The domains are defined according to eight demographic identities: male, female, LGBTQ, Christian, Muslim, other religions, Black, and White. All comments were randomly split into a disjoint training (269,038 comments), validation (45,180 comments), and test (133,782 comments) set.

Again, we follow Koh et al. (2021) and use a DistillBERT-base-uncased model with the following parameters: batch size = 16, learning rate = 1e-5, AdamW optimizer, number of epochs = 5, L2 regularization 0.01, and the maximum number of tokens of 300. We use these default parameters for training our DG layer. The performance is measured in the worst-group accuracy and we report mean and standard deviation across five runs.

**PovertyMap**   As the FMoW example shows, satellite images in combination with machine learning models can been used to monitor sustainability and economic challenges on a global scale. Another application of these satellite images is poverty estimation across different spatial regions. However, there exists a lack of labels for developing countries since obtaining the ground truth is expensive, which makes this application attractive for machine learning models. To study the OOD generalization to unseen countries, we use a modified version of the poverty mapping dataset of WILDS (Koh et al., 2021). The task is to predict a real-valued aset wealth index between 1 and 5 based on a multi-spectral satellite image. The domain refers to the country and whether the the the image is from a rural or urban are. In contrast to the other datasets, this dataset is split in five different folds, whereby in each fold the the training, valdiation and test set contains a disjoint set of countries, however, data from both rural and urban regions. The avergae size of each set across the 5 folds is for the training ~10,000 images (13-14 countries), ~4,000 images (4-5 different countries), and for the test set ~4,000 images (13-14 countries).

**On the challenge of obtaining domain labels.**   In the example of hospitals (e.g. Camelyon17 dataset), domain labels come, in fact, for free. However, other examples, such as the CivilComments dataset, show the opposite. This dataset requires additional annotations (i.e., demographic identities), which can be tedious to obtain in practice. Some algorithms need these domain annotations to achieve superior performance on each subgroup. Furthermore, the task of subgroup detection in itself is a difficult and relevant problem. Coming back to our hospital example, even people from the same hospital might belong to different subpopulation (e.g. gender, race, age) and these demographic subgroups are often more relevant for diagnosis than which hospital a patient comes from. This information, however, is not always available (due to anonymization standards, for instance) and, therefore, the relevant domain annotation might be hard to obtain.

We follow Koh et al. (2021) and use a pre-trained ResNet-18 model minimizing the sqarred error loss. For the optimization, we rely on the Adam optimizer with the following parameters: learning rate of 1e-3 with a decay of 0.96 per epoch, batch size of 64 and early stopping based on the OOD evaluation score. For evaluation, we report the Pearson correlation (r) between the predicted and actual asset wealth indices across the five different folds.

**General benchmark methods**   Following the WILDS benchmarking procedure (Koh et al., 2021), we compare our proposed DG layer to the following baselines. First, empirical risk minimization (ERM), which minimizes the average training loss over the pooled dataset. Second, a group of DG algorithms provided by the WILDS benchmark, namely, Coral, Fish, IRM, and DRO. The Coral algorithm introduces a penalty for differences in means and covariances of the domains feature distributions. The Fish algorithm achieves DG by approximating an inter-domain gradient matching objective, i.e., maximizing the inner product between gradients from different domains (Shi et al., 2021). Conceptually, Fish learns feature representations that are invariant across domains. Invariant risk minimization (IRM) introduces a penalty for feature distributions with different optimal classifiers for each domain (Arjovsky et al., 2019). The idea is to enable OOD generalization by learning domain-invariant causal predictors. Lastly, group distributionally robust optimization (DRO) explicitly minimizes the training loss on the worst-case domain (Sagawa et al., 2020; Hu et al., 2018).

In addition to the baselines originally presented in Koh et al. (2021), we consider the following more recent DG baselines. First, we describe LISA, which instead of regularizing the internal representations for generalization, seeks to learn domain-invariant predictors with selective data

Table 13: Parameters for DG Layer in WILDS experiments for the Fine Tuning (FT) and End-to-end training (E2E) Settings.

| | EXPERIMENT | | M | N | $\lambda_{L_1}$ | $\lambda_{OLS}$ | $\lambda_{ORTH}$ | $\sigma$ | $\kappa$ |
|---|---|---|---|---|---|---|---|---|---|
| WILDS BENCHMARK | FT AND E2E | CS | 5 | 10 | $1e^{-3}$ | $1e^{-3}$ | - | 4 | 2 |
| | | MMD | 5 | 10 | $1e^{-3}$ | $1e^{-3}$ | - | 4 | 2 |
| | | PROJECTION | 5 | 10 | - | $1e^{-3}$ | $1e^{-3}$ | 16 | - |

augmentation Yao et al. (2022). Common Gradient Descent (CGD), introduced by Piratla et al. (2021), is based on Group-DRO. However, it proposes to focus not on groups with the worst regularization but on common groups that enable generalization. Last, Adaptive Risk Minimization using batch normalization (ARM-BN) by Zhang et al. (2021) is different from the methods presented since it adapts to previously unseen domains during test time using unlabeled observations from this test domain.

Table 14: Detailed results on RxRx1 dataset.

| | | RxRx1 | | |
| | | VAL ACCURACY | IID ACCURACY | OOD ACCURACY |
|---|---|---|---|---|
| **ERM** | | 19.4 (0.2) | 35.9 (0.4) | 29.9 (0.4) |
| **CORAL** | | 18.5 (0.4) | 34.0 (0.3) | 28.4 (0.3) |
| **FISH** | | - | - | |
| **IRM** | | 5.6 (0.4) | 9.9 (1.4) | 8.2 (1.1) |
| **GROUP DRO** | | 15.2 (0.1) | 28.1 (0.3) | 22.5 (0.3) |
| **LISA** | | 20.1 (0.4) | 41.1 (1.3) | 31.9 (1.0) |
| **CGD** | | - | - | - |
| **ARM-BN** | | 20.9 (0.2) | 34.9 (0.2) | 31.2 (0.1) |
| | CS | 18.9 (0.4) | 36.0 (0.4) | 29.7 (0.4) |
| **FT** | MMD | 19.0 (0.2) | 36.0 (0.2) | 29.6 (0.2) |
| | PRO | 18.5 (2.5) | 35.1 (0.3) | 29.0 (0.2) |
| | CS | 19.5 (0.5) | 36.2 (0.4) | 29.9 (0.3) |
| **E2E** | MMD | 19.5 (0.5) | 36.2 (0.4) | 29.9 (0.3) |
| | PRO | 19.3 (0.5) | 36.0 (0.5) | 29.8 (0.3) |

Table 15: Detailed results on FMoW dataset.

| | | FMoW | | | |
| | | VAL | | TEST | |
| | | AVG ACC | WORST-REGION ACC | AVG ACC | WORST-REGION ACC |
|---|---|---|---|---|---|
| **ERM** | | 59.2 (0.07) | 49.8 (0.36) | 52.7 (0.23) | 31.3 (0.17) |
| **CORAL** | | 56.5 (0.15) | 48.9 (1.31) | 50.1 (0.07) | 32.8 (0.66) |
| **FISH** | | 57.8 (0.15) | 49.5 (2.34) | 51.8 (0.32) | 34.6 (0.18) |
| **IRM** | | 56.1 (0.61) | 49.7 (0.97) | 50.4 (0.75) | 32.8 (2.09) |
| **GROUP DRO** | | 57.6 (0.70) | 48.7 (0.92) | 52.8 (1.15) | 31.1 (1.66) |
| **LISA** | | 58.7 (1.12) | 48.7 (0.92) | 52.8 (1.15) | 35.5 (0.81) |
| **CGD** | | 57.0 (1.03) | 49.8 (1.04) | 50.6 (1.39) | 32.0 (2.26) |
| **ARM-BN** | | 48.0 (0.65) | 38.9 (2.17) | 42.1 (0.26) | 24.4 (0.54) |
| | CS | 59.6 (0.21) | 50.4 (0.50) | 53.1 (0.22) | 31.8 (1.24) |
| **FT** | MMD | 59.6 (0.24) | 50.3 (0.60) | 53.1 (0.22) | 31.9 (1.17) |
| | PRO | 59.2 (0.36) | 50.0 (0.76) | 52.7 (0.14) | 31.8 (1.08) |
| | CS | 59.3 (0.33) | 52.1 (0.80) | 53.4 (0.25) | 34.4 (1.86) |
| **E2E** | MMD | 58.5 (0.41) | 52.9 (1.67) | 52.7 (0.45) | 34.4 (0.71) |
| | PRO | 58.5 (0.43) | 50.9 (0.85) | 52.7 (0.68) | 32.9 (0.78) |

