# OpenReview forum: "Gated Domain Units for Multi-source Domain Generalization"
_ICLR.cc/2023/Conference — Submitted to ICLR 2023_

### Official Review · Reviewer_13Xi · 2022-10-19

**Confidence:** 4
**Correctness:** 3
**Technical Novelty And Significance:** 3
**Empirical Novelty And Significance:** 2
**Recommendation:** 5

**Clarity, Quality, Novelty And Reproducibility:**

I found the idea of introducing the I.E.D. assumption (i.e. all domains are a mixture of the same elementary distributions) novel and potentially promising. The algorithmic contributions to leverage the I.E.D. assumption when training deep neural networks for domain generalization problems is interesting and has a solid motivation. However, as I mentioned in the “Strengths and Weaknesses“ section, I have multiple concerns with the current version of the manuscript, especially regarding the soundness of the theoretical motivation for the I.E.D. assumption and the empirical evaluation of the proposed approach. In the following, I detail my concerns and provide suggestions for the authors:

- It is not clear to me why Lemma 1 would show the theoretical appeal of the I.E.D. assumption. As far as I understood, Lemma 1 only shows that the Bayes predictor f^* will be Pareto optimal in case the I.E.D. assumption holds, which basically means that f^* can, for example, be such that only of the risks is minimized (i.e. f^* corresponds to a solution in one of the extremes of the Pareto front). I find this actually not appealing from the perspective of domain generalization, since solutions in the extremes of the Pareto front could be enforcing the predictor to yield unfair predictions in cases where, for example, each elementary unit corresponds to a subgroup.

- A standard assumption in domain generalization is that of covariate shift [1] (in summary, conditional labeling distribution is the same across all domains). However, it is not clear whether this assumption is required for the use of GDUs to make sense.

- I disagree with the claim that obtaining domain information is challenging, especially in the practical cases considered in this manuscript where, for most of the datasets taken into account, domain labels come “for free” in the data collection process (i.e. different domains correspond to data collected from different hospitals).

- The manuscript focused too much in comparing the introduced approach with other methods rather than deeply understanding the role of each introduced component. Also, even though the experiments may show that GDUs are helpful for improving out-of-distribution generalization, they don't show whether this is happening due to what is claimed in the manuscript (i.e. the I.E.D. assumption indeed holds in practice and GDUs are in fact capable of learning to model the elementary distributions). Therefore, experiments that verify whether the proposed improvements are responsible for the reported increase in performance should be included in the main paper.

- The manuscript lacks discussions about how not satisfying the I.E.D. assumption would affect the predictions of models trained with their proposed approach, as well as to which (unseen) distributions it is possible to expect generalization given that the I.E.D. assumption holds. One way to address that could be to include results that shed light on how much the performance of a trained ensemble containing GDUs would decrease in case a test domain presents no common elementary distribution with respect to the training domains.

- It is not clear from the experiments (including the ones reported in the Appendix) what is the impact of the choice of the number of elementary units. Since the authors do not provide any insight from theory, i.e. there are no results that ties out-of-distribution generalization with learning the “correct“ amount of elementary distributions, at least experiments studying this factor should be provided.

- The domain generalization setting requires an ability to generalize in- and out-of-distribution. How well do the models reported perform in-distribution?

- Selected baselines are not the approaches more closely related to the proposed algorithm. Why exactly were those baselines selected? Why haven’t the authors compared GDUs with the most related work to their approach [2, 3] which was cited in the Related Work section?



Clarity: overall, I found the manuscript well-written, but the points need to be addressed in order to improve its clarity:
- The manuscript contains too many acronyms, which should be avoided as they make it difficult to understand some sentences in the text.
- Some sentences and terms across the manuscript are unclear:
  - Page 1, paragraph 2: it is not clear what "smaller unit" means in the context of this work. As far as I understood, a "unit" would be a distribution, but what is a "small" distribution? Please clarify.
  - Page 2, Section 2.1: “The advantage is that we can find an invariant subspace at a more elementary level”. What exactly does “elementary level” mean in this sentence?
  - Page 2, paragraph 2: “The question arises if and when elementary domains evolve.” Please clarify what is the meaning of “evolve” here.

[1] David, Shai Ben, et al. "Impossibility theorems for domain adaptation", 2010. \
[2] Monteiro, Joao, et al. "Domain Conditional Predictors for Domain Adaptation", 2021. \
[3] Piratla, Vihari, et al "Efficient domain generalization via common-specific low-rank decomposition", 2020.


**Strength And Weaknesses:**

Strengths:
- The paper tackles a relevant and open problem for the machine learning community;
- The proposed approach is theoretically grounded and its algorithmic instantiation seems practical. Also, it does not rely on domain labels;
- Empirical evaluation was extensive in the sense that multiple datasets and tasks were taken into account;
- Results show that the introduced method presents advantages in comparison to the selected baselines in terms of performance on unseen domains at training time.

Weaknesses:
- Lack of soundness in the theoretical motivation, more specifically in Lemma 1 (see the following section of the review for more details);
- The manuscript contains multiple claims and statements that are not well-supported / clear (see the following section of the review for more details);
- It is not clear, from both theoretical and empirical perspectives, what is the impact on out-of-distribution generalization of selecting a much higher / lower number of elementary distributions;
- Experimental evaluation lacks in many aspects:
  - In the main paper, the proposed approach is only evaluated in terms of performance on unseen domains. This type of evaluation does not explain the actual source of the observed improvements;
  - The authors did not take into account closely related approaches as baselines in the empirical evaluation.


**Summary Of The Paper:**

This contribution tackles the domain generalization problem by assuming that every real-world distribution is composed of a mixture of elementary distributions, which remain invariant across different domains. This assumption was named I.E.D. (Invariant Elementary Distribution). The authors presented a lemma to support the theoretical soundness of making such an assumption and then introduced a practical way to leverage the I.E.D. assumption when training neural networks to enable generalization on unseen domains containing the same elementary distributions. The proposed approach roughly consists of an ensemble of predictors weighted by the similarity between an instance and each such elementary distribution. The main contribution of this work lies in computing such similarities with the introduced Gated Domain Unit (GDU), a module to be employed along with neural networks, responsible for encoding each elementary distribution into an embedding that can be trained via backpropagation with the parameters of the ensemble. Empirical evaluation of the proposed approach was carried out on multiple benchmarks comprising, for example, computer vision tasks and time-series classification. Results showed the proposed approach outperforms the selected baselines with respect to accuracy on unseen domains at training time.

**Summary Of The Review:**

This work tackles the domain generalization setting and assumes that problems within this setting can be seen as if each domain is a mixture of the same distributions (elementary distributions). The authors attempted to provide in Lemma 1 a theoretical motivation to support the introduced assumption, but I have concerns and questions about this result.  The authors then proposed architectural changes to deep neural networks by introducing Gated Domain Units to allow generalization to unseen domains for which the introduced assumption holds. Even though I appreciate the practical relevance of the problem being tackled by this submission, I found that the manuscript lacks in multiple points (please refer to the previous sections of this review for comments and suggestions), especially in the motivation for the introduced assumption and empirical evaluation in terms of considered baselines, reported metrics, as well as the aspects investigated in the experiments. All in all, I believe this submission requires multiple improvements in order to be considered for publication.

---

### Official Review · Reviewer_FZYk · 2022-10-26

**Confidence:** 3
**Correctness:** 3
**Technical Novelty And Significance:** 3
**Empirical Novelty And Significance:** 3
**Recommendation:** 6

**Clarity, Quality, Novelty And Reproducibility:**

The paper is easy to read
Ablation studies are good.
The extensive supplementary material and the model diagrams help reproducibility.

**Strength And Weaknesses:**

The idea is theoretically interesting and it certainly is versatile within the convex hull of the base domains. It relates to some of the existing work on recycling neural networks. I suggest the authors to cite those papers and if possible compare with those baselines.


Some questions for the authors:
- in figure 7 of the supplementary material, how come adding elementary domains does not improve results?
- The base domains are based on the domains observed from source data. Therefore we can only assume the generalizability of this approach extends to test domains which can be considered an interpolation of source domains. How can the generalizability of the approach extrapolate outside the convex hull of the source domains?



**Summary Of The Paper:**

The paper presents a method for domain generalization. The core assumption is that every domain can be decomposed to some elementary domain (something like a base domains) which are invariant. Every domain is a linear combination of these base domains. With this assumption, we can have elementary domain prediction functions and we just need to figure out how to linearly combine their predictions. For a given input example, this is done by measuring the similarity of the input example with each base domain.

**Summary Of The Review:**

The paper is theoretically interesting. Im not sure how applicable it is to practical applications for example generalizability outside of the convex hull of the source domains.

---

### Official Review · Reviewer_1an7 · 2022-10-26

**Confidence:** 3
**Correctness:** 3
**Technical Novelty And Significance:** 2
**Empirical Novelty And Significance:** 2
**Recommendation:** 3

**Clarity, Quality, Novelty And Reproducibility:**

I find it hard to judge the novelty of the method, given that it is mainly based on the I.E.D assumption. We need more discussion in the rebuttal to understand the assumption.

**Strength And Weaknesses:**

In my opinion, the main weaknesses of the paper are the assumption of I.E.D (which the authors did discuss briefly) and the evaluation of the method. Details are in the section below.

**Summary Of The Paper:**

Based on an assumption about invariant elementary distributions, the authors proposed a method, namely Gated Domain Units, to recover those elementary distributions and leverage them for the domain generalization problem.

**Summary Of The Review:**

I would like some clarification on the following issues:
- The authors claim that "Lemma 1 shows the theoretical appeal of the I.E.D. assumption". Can the authors elaborate? Even if we know the I.E.D.s and their coefficient in the mixture (which we would never know, but just assume here), then still it is very hard to find the set of Pareto optimal solution $f$. Then why Lemma 1 would show the theoretical appeal of this assumption or motivate us to find such elementary distributions?

- The practicality of the I.E.D. assumption: can the authors give an example of the I.E.D. in some of the datasets used in the paper? The authors give one example in section 2.1 but I don't think it is concrete enough. For me, an ideal way to do this for 2 domain $p_s$, $p_t$ is to somehow decompose them into $p_s = \alpha_s^1.p_s^1 + ... + \alpha_s^k.p_s^k$ and $p_t = \alpha_t^1.p_t^1 + ... + \alpha_t^k.p_t^k$ such that a discriminator can't distinguish between $p_s^i$ and $p_t^i$ well.

- The empirical evaluation is also not quite convincing to me. The method needs to process data with multiple Gated Domain Units, thus increasing the training and inference complexity. Therefore, ensemble is a natural baseline for this method. And indeed, the authors do include this baseline for mnist and ECG. But I wonder why the authors do not consider this baseline for WILDS? This would make the empirical evaluation much stronger.

---

### Decision · Program_Chairs · 2023-01-20

**Decision:**

Reject

**Justification For Why Not Higher Score:**

There are concerns about lack of soundness in the theoretical motivation, lack of clarity about the impact the number of elementary distributions on out-of-distribution generalization, and lack of comparison with related approaches in the literature.

**Justification For Why Not Lower Score:**

N/A

**Metareview: Summary, Strengths And Weaknesses:**

The paper proposes a method to address the domain generalization problem by assuming that real-world distributions are composed of elementary distributions that remain invariant across different environments. This assumption is named I.E.D. (Invariant Elementary Distribution). The authors present a lemma that shows the IED assumption implies invariant structure in the solution space. Authors propose an approach that roughly consists of an ensemble of predictors weighted by the similarity between an instance and each such elementary distribution. Empirical results on computer vision tasks and time-series classification are presented to show the effectiveness of the proposed method. Reviewers acknowledge that the paper tackles a relevant and open problem, and the proposed approach is theoretically grounded. However there are concerns about lack of soundness in the theoretical motivation, lack of clarity about the impact the number of elementary distributions on out-of-distribution generalization, and lack of comparison with related approaches in the literature. The IED assumption is interesting and intuitive, and the meta-reviewer hopes that authors will use the reviewers' feedback to strengthen the paper.